# TOWARDS SEED-INVARIANT SAFETY ALIGNMENT IN TEXT-TO-IMAGE MODELS

## ABSTRACT

Text-to-image diffusion models have achieved remarkable success in generating high-quality images, yet existing safety mechanisms exhibit critical *cross-seed instability* where defense performance varies significantly under different random seed conditions. This instability stems from the fact that a single malicious prompt generates diverse harmful variants across different noise initializations, forming complex distributional clusters that current methods cannot adequately address. We investigate extending Noise Contrastive Alignment (NCA) to diffusion models due to its native capability of handling multiple negative samples through probabilistic weighting, but our theoretical analysis reveals two fundamental flaws in direct extension: *gradient reversal* caused by positive regularization terms that paradoxically penalize safe content generation, and *uniform suppression* of harmful samples that ignores severity variations. To tackle these issues, we propose *Noise Contrastive Diffusion* (NCD), which incorporates targeted algorithmic modifications including elimination of problematic regularization and introduction of pairwise regularization mechanisms that establish individualized preference relationships between safe and harmful variants. Extensive experiments further demonstrate that NCD achieves superior cross-seed stability, reducing attack success rates (ASRs) from 11.1% to 6.2% compared to SOTA methods while maintaining exceptional generation quality, exhibiting robust resistance against sophisticated jailbreak prompts and strong generalizability across different T2I architectures.
**WARNING: This paper may contain examples of harmful texts and images.**

## 1 INTRODUCTION

Recently, driven by great improvements in model architecture and advancements in semantic understanding techniques (Saharia et al., 2022; Ramesh et al., 2022; Podell et al., 2023; Esser et al., 2024), text-to-image (T2I) models have become capable of generating high-quality images with remarkable fidelity to user instructions. Although this technique exhibits remarkable capabilities for content creation (Peebles & Xie, 2023; Zhang et al., 2023) and artistic rendering (Ruiz et al., 2023; Wang et al., 2024), it also poses several risks. Since their trainings rely on vast, uncurated internet data, T2I models can be exploited to produce harmful imagery depicting sexual (Wen et al., 2024), violent (Schramowski et al., 2023), or biased content (Friedrich et al., 2023). This escalating concern has compelled the research community to prioritize the safe output of T2I models.

To mitigate the generation of harmful content, current research has focused on applying rigorous safety mechanisms to T2I models. External filter-based methods (CompVis, 2023; Khader et al., 2024; Huggingface, 2025) employ post-hoc detection of harmful content through dedicated classifiers, while training-free approaches (Schramowski et al., 2023; Yoon et al., 2024) modulate generation behavior during inference without parameter modification. However, filter-based methods offer suffer from limited robustness against jailbreak attacks (Huang et al., 2024b), and simultaneously, training-free methods require careful hyperparameter tuning and may compromise overall generation quality. Due to such limitations, increasing focus has shifted toward parameter modification approaches that directly alter model weights for more robust and permanent safety guarantees. For instance, concept erasing methods (Kumari et al., 2023; Gandikota et al., 2023; Zhang et al., 2024a) train models to forget inappropriate concepts, while model editing techniques (Gandikota et al.,

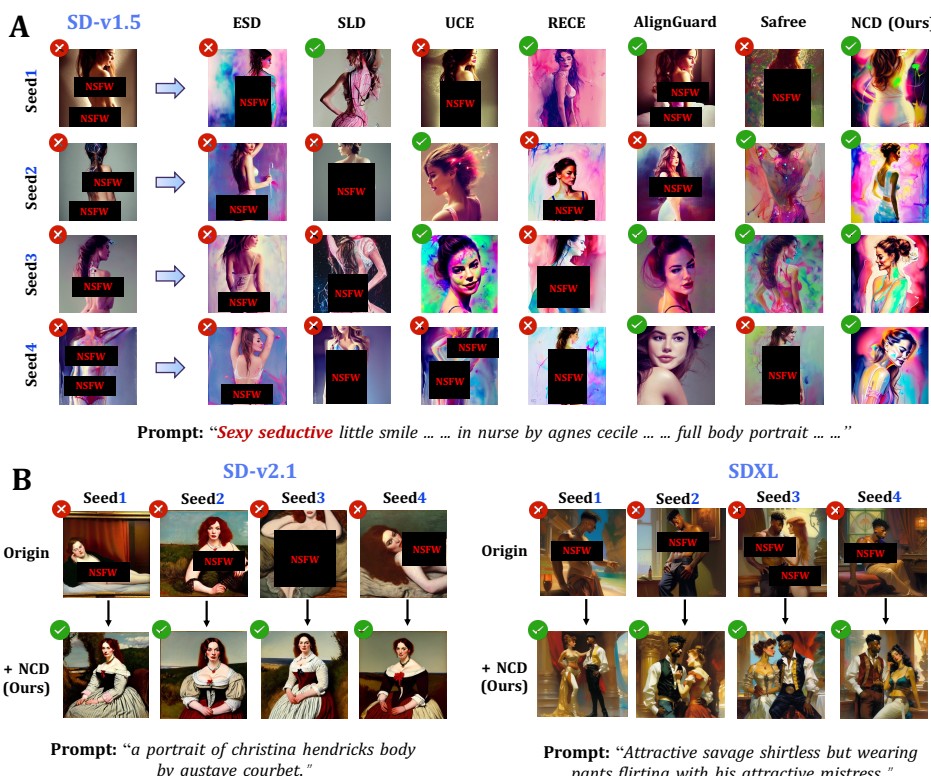

Figure 1: We propose NCD, a multi-objective preference calibration framework for safety alignment that effectively mitigates the defense vulnerabilities of diffusion model safety mechanisms when facing malicious prompts across multiple noise seed variants. *(A)* Defense performance comparison of various mechanisms under different random seeds for harmful inputs. NCD consistently defends against all noise variants while other methods show vulnerability. *(B)* Demonstration of NCD framework's generalizability on additional models including SD-v2.1 and SDXL.

2024; Gong et al., 2024) achieve targeted interventions by modifying attention projection matrices to redirect harmful embeddings toward safe alternatives.

However, despite demonstrating effectiveness in mitigating harmful content generation, existing safety mechanisms exhibit significant performance variations under different random seed conditions, as illustrated in Fig 1. This *cross-seed instability* exposes a fundamental limitation: current approaches fail to establish robust safety alignment that can consistently address the diverse harmful variants produced by malicious prompts across different noise initializations.

Given the one-to-many nature of the cross-seed instability challenge, Noise Contrastive Alignment (NCA) (Chen et al., 2024) emerges as a particularly suitable framework. Specifically, through extending NCA to diffusion models, we could simultaneously address diverse harmful variants generated from different seeds within a unified optimization framework. However, through rigorous theoretical analysis and verification, we identify that the direct extension of NCA to diffusion mode introduces two fundamental algorithmic flaws that severely compromise training stability and alignment effectiveness: First, the positive regularization term in the original NCA objective exhibits a critical ❶ *gradient reversal* pathology: as safety alignment progresses and rewards for safe content improve, the regularization term paradoxically begins to dominate gradient updates, causing the optimization to penalize rather than promote safe content generation. Furthermore, we demonstrate that NCA's ❷ *uniform suppression strategy for harmful samples* creates a severe mismatch with the inherent diversity of cross-seed harmful content, where samples exhibit varying degrees of severity yet receive identical optimization signals, preventing the model from developing nuanced discriminative capabilities across different manifestations of harmful content.

To address these inherent issues, we propose Noise Contrastive Diffusion (NCD), a novel multi-seed alignment framework that incorporates targeted algorithmic modifications to overcome the limitations of direct NCA extension. Specifically, our NCD eliminates the problematic positive regularization term to ensure consistent gradient directions for safe content optimization while introducing pairwise regularization mechanisms that establish individualized preference relationships between safe samples and each harmful variant. Extensive experimental validation further demonstrates that NCD could achieve exceptional cross-seed stability, reducing ASRs from 11.1% to 6.2% compared to RECE while maintaining superior generation quality. Additionally, against sophisticated jailbreak prompts, NCD exhibits robust defense capabilities, achieving a 5.0% ASR on Sneaky-Prompt (SP) compared to 10.5% for the state-of-the-art method AlignGuard. Moreover, NCD demonstrates strong generalizability across T2I architectures, consistently outperforming existing methods on both SD v2.1 and SDXL with significant performance improvements across all safety metrics.

## 2 RELATED WORKS

### 2.1 TEXT-TO-IMAGE (T2I) GENERATION

Owing to more stable training dynamics and improved generation fidelity, diffusion models have gained prominence over early GAN-based approaches (Goodfellow et al., 2020; Esser et al., 2021). This transformation begins with Diffusion Probabilistic Models (DDPM) (Ho et al., 2020), which establishes the foundational denoising framework. Subsequently, Latent Diffusion Models(Rombach et al., 2022) advance the paradigm by operating in latent space and incorporating classifier-free guidance (Ho & Salimans, 2022), thereby achieving improved computational efficiency and text-image alignment. Building on these foundations, contemporary state-of-the-art models, including Imagen (Saharia et al., 2022), DALL·E (Ramesh et al., 2021; 2022), and the Stable Diffusion series (Rombach et al., 2022; Podell et al., 2023; Esser et al., 2024), showcase the remarkable capabilities of large-scale diffusion architectures in generating photorealistic images from textual descriptions. Moreover, recent developments in human preference alignment, such as Diffusion-DPO (Wallace et al., 2024), D3PO (Yang et al., 2024a), and DSPO (Zhu et al., 2025), further enhance these models through reinforcement learning from human feedback. However, alongside their impressive generation capabilities, these powerful T2I models introduce significant safety concerns, including the potential for harmful content generation, thereby necessitating robust defense mechanisms.

### 2.2 SAFETY MECHANISMS IN T2I MODELS

Ensuring safety and adherence to ethical norms in generation has become a critical issue in Text-to-Image (T2I) models, with existing approaches falling into three main paradigms. *Filter-based methods* serve as external safety mechanisms that detect harmful content through textual filtering (CompVis, 2023; Khader et al., 2024), LLM determination (Markov et al., 2023), or image analysis (Rombach et al., 2022). For instance, Stable Diffusion's safety checker computes cosine similarity between generated images and predefined harmful concept embeddings to reject unsafe outputs. However, these external approaches suffer from limited robustness against adversarial attacks (Rando et al., 2022; Yang et al., 2024d) and thus can be easily bypassed by advanced jailbreak prompts. *Training-free methods* offer an intermediate solution that modulates generation behavior during inference. For example, SLD (Schramowski et al., 2023) employs classifier-free safety guidance by incorporating negative prompts during the denoising process, while Safree (Yoon et al., 2024) identifies toxic concept subspaces in text embedding space and steers prompt embeddings away from these harmful regions. Due to the need for more robust and permanent safety guarantees, recent research has increasingly focused on *parameter modification approaches* that directly alter model weights to suppress harmful concept generation from within the model itself. Supervised methods like ESD (Gandikota et al., 2023) and CA (Kumari et al., 2023) train models to "forget" specific concepts, while attention-based techniques such as Forget-Me-Not (Zhang et al., 2024a) fine-tune cross-attention layers to redirect attention away from harmful content. Model editing methods, including UCE (Gandikota et al., 2024) and RECE (Gong et al., 2024) achieve more targeted interventions by directly modifying cross-attention projection matrices through closed-form solutions.

Despite these advances, existing methods struggle to balance generation quality with safety and lack systematic defense against input noise variations, leading to inconsistent protection across different

random seeds. In contrast, our proposed NCD framework addresses these limitations through multi-noise contrastive alignment and preference calibration optimization, which we detail in Secs 3 and 4.

## 3 SEED-INVARIANT SAFETY ALIGNMENT VIA NCA

### 3.1 PRELIMINARIES

**Direct Preference Optimization for Diffusion Models.** Direct Preference Optimization (DPO) is a contrastive learning framework that aligns large language models with human preferences without requiring explicit reward model training. The algorithm leverages paired preference data consisting of preferred samples $y^w$ and rejected samples $y^l$, optimizing the following objective:

$$\mathcal{L}_{\text{DPO}}(p_\theta; p_{\text{ref}}) = -\mathbb{E}_{(x,y^w,y^l)\sim\mathcal{D}} \left[ \log \sigma \left( r_\theta(x, y^w) - r_\theta(x, y^l) \right) \right], \tag{1}$$

where $x$ denotes the input prompt, $r_\theta(x, y) = \beta \log \frac{p_\theta(y|x)}{p_{\text{ref}}(y|x)}$, $\sigma(\cdot)$ is the sigmoid function, $p_\theta$ is the policy being optimized, $p_{\text{ref}}$ is the reference policy, and $\beta$ controls the KL regularization strength. The implicit reward function $r(x, y) = \beta \log \frac{p_\theta(y|x)}{p_{\text{ref}}(y|x)}$ encourages the model to increase the likelihood of preferred outputs relative to rejected ones while maintaining proximity to the reference.

Recently, DPO has been successfully extended to diffusion models for image generation tasks. Diffusion-DPO adapts this framework to align text-to-image diffusion models with human preferences. Given a text prompt $c$ and corresponding preferred image $x^w$ and rejected image $x^l$, the objective function is formulated as:

$$\mathcal{L}_{\text{Diff-DPO}}(\theta) = -\mathbb{E}_{t\sim\mathcal{U}(1,T)} \left[ \log \sigma \left( R_\theta(c, x_t^w) - R_\theta(c, x_t^l) \right) \right], \tag{2a}$$

$$R_\theta(c, x_t) = K \cdot \left[ \mathcal{L}_{\text{diff}}(\epsilon_{\text{ref}}, x_t, c, t) - \mathcal{L}_{\text{diff}}(\epsilon_\theta, x_t, c, t) \right], \tag{2b}$$

where $\epsilon_\theta$ denotes the denoising network being optimized, $\epsilon_{\text{ref}}$ represents the frozen reference network, $K > 0$ is a scaling hyperparameter, and $t \sim \mathcal{U}(1, T)$ indicates uniform sampling over diffusion timesteps. The diffusion loss $\mathcal{L}_{\text{diff}}(\epsilon, x_t, c, t) = \|\tilde{\epsilon}_t - \epsilon(x_t, c, t)\|^2$ measures the mean squared error between the ground-truth noise $\tilde{\epsilon}_t$ and the network's prediction. The step-wise implicit reward $R(c, x_t) = K \cdot \Delta\mathcal{L}_t$ is determined by the relative improvement in denoising performance, encouraging the diffusion model to generate images that align with human preferences by maximizing the likelihood of preferred samples while reducing that of rejected samples.

**Noise Contrastive Alignment (NCA).** NCA (Chen et al., 2024) is an alignment framework based on Noise Contrastive Estimation (NCE) that addresses the limitation of DPO methods which can only handle pairwise preference data. NCA reformulates the language model alignment problem as a multi-choice binary classification task: given candidate responses $\{y_1, y_2, \ldots, y_N\}$ for input $x$, the model learns by predicting the probability of sampling the corresponding response from the target policy $\pi_\theta$ as $p_\theta(\nu = 1|x, y_i) = \sigma(r_\theta(x, y_i))$. By maximizing the likelihood estimation of these probabilities, NCA constructs the following objective:

$$\mathcal{L}_{\text{NCA}}(\theta) = -\mathbb{E}_{(x,\{y_i,r_i\}_{1:N})\sim\mathcal{D}} \left[ \sum_{i=1}^{N} w_i \log \sigma\left(r_\theta(x, y_i)\right) + \frac{1}{N} \sum_{i=1}^{N} \log \sigma\left(-r_\theta(x, y_i)\right) \right], \tag{3}$$

where $w_i = \frac{e^{r_i/\alpha}}{\sum_{j=1}^{N} e^{r_j/\alpha}}$ represents the softmax-normalized weight based on explicit reward $r_i$, $\alpha > 0$ is the temperature parameter, and $r_\theta(x, y) = \beta \log \frac{p_\theta(y|x)}{p_{\text{ref}}(y|x)}$ is the implicit reward function. The first term encourages high-quality responses based on their reward-weighted importance, while the second term provides contrastive learning by treating all candidates as negative samples with equal weight. Unlike DPO, which primarily focuses on adjusting relative likelihood across different responses, NCA optimizes the absolute likelihood of each response, effectively preventing the likelihood degradation of preferred responses that commonly occurs in DPO training.

### 3.2 EXTENDING NCA TO MULTI-SEED SAFETY ALIGNMENT (A DIRECT EXTENSION)

The NCA framework's ability to simultaneously handle multiple preference samples and optimize through probabilistic weighting aligns well with the requirements of multi-seed safety alignment. Building upon NCA, we develop a multi-seed safety alignment method for diffusion models.

In the first, considering the multi-step generation nature of diffusion models, we follow Diffusion-DPO by replacing the original policy distribution with the diffusion model's conditional distribution $p(x_{0:T}|c)$ and implicitly expressing the reward function through the ratio of conditional distributions. The corresponding objective function of NCA becomes:

$$\mathcal{L}_\theta = -\mathbb{E}\left[\sum_{i=1}^N \left(w_i \log \sigma\left(r_\theta(c, x^i)\right) + \frac{1}{N}\log \sigma\left(-r_\theta(c, x^i)\right)\right)\right], \tag{4}$$

where $w_i = \frac{e^{s_i/\alpha}}{\sum_{j=1}^N e^{s_j/\alpha}}$ represents the softmax weight of the $i$-th sample based on safety score $s_i$, and $r_\theta(c, x^i) = \beta\mathbb{E}\left[\log\frac{p_\theta(x_{0:T}^i|c)}{p_{\text{ref}}(x_{0:T}^i|c)}\right]$ denotes the implicit reward for the whole denoising trajectory. However, optimizing over entire trajectories is computationally expensive. Following the upper bound derivation in Diffusion-DPO, we extend their Jensen's inequality-based approach to our multi-seed weighted setting. Under the Markov property of the diffusion process and applying Jensen's inequality, we obtain:

$$\mathcal{L}_\theta \leq -\mathbb{E}_t\left[\sum_{i=1}^N \left(w_i \log \sigma\left(\tilde{r}_\theta(c, x_t^i)\right) + \frac{1}{N}\log \sigma\left(-\tilde{r}_\theta(c, x_t^i)\right)\right)\right], \tag{5}$$

where $\tilde{r}_\theta(c, x_t^i) = \beta T \log\frac{p_\theta(x_t^i|x_{t+1}^i, c)}{p_{\text{ref}}(x_t^i|x_{t+1}^i, c)}$ represents the step-wise reward approximation. This extension enables efficient step-wise optimization with theoretical guarantees.

To make this practically computable, we then leverage the DDPM parameterization. Under DDPM, the reverse process can be approximated by the posterior distribution $q(x_{t-1}|x_t, x_0)$, enabling the transformation of log probability ratios into KL divergences at each timestep. The KL divergence terms can be expressed as mean squared loss of noise prediction, yielding the simplified objective:

$$\mathcal{L}(\theta) = -\mathbb{E}_t\left[\sum_{i=1}^N \left(w_i \log \sigma\left(R_\theta(c, x_t^i)\right) + \frac{1}{N}\log \sigma\left(-R_\theta(c, x_t^i)\right)\right)\right], \tag{6}$$

where $R_\theta(c, x_t^i) = K\left(\mathcal{L}_{\text{diff}}(\epsilon_{\text{ref}}, x_t^i, c) - \mathcal{L}_{\text{diff}}(\epsilon_\theta, x_t^i, c)\right)$ represents the step-wise implicit reward, and $\mathcal{L}_{\text{diff}}(\epsilon, x_t, c) = \|\tilde{\epsilon}_t - \epsilon(x_t, c)\|^2$ is the standard denoising loss.

Finally, we reformulate the loss function into an explicit preference form. Specifically, for each malicious prompt, we have one safe response $x^w$ (preferred) and multiple harmful responses $\{x^{l_j}\}_{j=1}^{N-1}$ (rejected) generated with different random seeds. The safety-oriented objective becomes:

$$\begin{aligned}
\mathcal{L}(\theta) = -\mathbb{E}_t\Bigg[&\mathbf{w}^w \log \sigma\left(R_\theta(x_t^w)\right) + \frac{1}{N}\log \sigma\left(-R_\theta(x_t^w)\right) \\
&+ \sum_{j=1}^{N-1}\left(w^{l_j} \log \sigma\left(R_\theta(x_t^{l_j})\right) + \frac{1}{N}\log \sigma\left(-R_\theta(x_t^{l_j})\right)\right)\Bigg],
\end{aligned} \tag{7}$$

where $R_\theta(x_t^w)$ and $R_\theta(x_t^{l_j})$ represent step-wise rewards for the safe and $j$-th harmful samples, respectively, and $\mathbf{w}^w$, $w^{l_j}$ denote their corresponding importance weights. In practice, we set $\mathbf{w}^w \approx 1$ and $w^{l_j} \approx -1$ to enforce safety alignment during training.

## 3.3 POTENTIAL ISSUES OF DIRECT EXTENSION

While models trained following the above paradigm successfully mitigate harmful content generation across multiple random seed settings, our practical implementation reveals two fundamental limitations when directly adapting NCA algorithms to diffusion models. These limitations arise from the intrinsic design differences between NCA and the safety finetuning requirements.

### 3.3.1 REVERSE UPDATE INDUCED BY POSITIVE REGULARIZATION TERM

In the direct extension, the $1/N$ regularization term is designed to collaborate with the optimization objective in jointly determining the loss update direction. However, upon direct application to diffusion model safety alignment tasks, we identify a critical issue that undermines training stability.

**Theorem 3.1.** *(Gradient Reversal) For the safety-oriented diffusion loss $\mathcal{L}(\theta)$ defined in Equation (7), the gradient component for safe samples with weight $w^w \approx 1$ is given by:*

$$\nabla_\theta \mathcal{L}^w = -\mathbb{E}_t \left[ \left( 1 - \frac{N+1}{N} \sigma(R_\theta(x_t^w)) \right) \nabla_\theta R_\theta(x_t^w) \right], \tag{8}$$

*When $\sigma(R_\theta(x_t^w)) > \frac{N}{N+1}$, the gradient coefficient becomes negative, causing reverse update behavior that reduces the likelihood of generating safe content.*

*Proof.* The diffusion loss for safe samples $\mathcal{L}^w$ has the following form:

$$\mathcal{L}^w(\theta) = -\mathbb{E}_t \left[ \mathbf{w}^w \log \sigma(R_\theta(x_t^w)) + \frac{1}{N} \log \sigma(-R_\theta(x_t^w)) \right]. \tag{9}$$

We directly calculate the gradient with respect to $\theta$:

$$\nabla_\theta \mathcal{L}^w = -\mathbb{E}_t \left[ \left( \mathbf{w}^w - (\mathbf{w}^w + \frac{1}{N}) \sigma(R_\theta(x_t^w)) \right) \nabla_\theta R_\theta(x_t^w) \right]. \tag{10}$$

Since the importance weight for safe samples $\mathbf{w}^w \approx 1$, the above equation simplifies to:

$$\nabla_\theta \mathcal{L}^w = -\mathbb{E}_t \left[ \left( 1 - \frac{N+1}{N} \sigma(R_\theta(x_t^w)) \right) \nabla_\theta R_\theta(x_t^w) \right]. \tag{11}$$

This theorem reveals a fundamental flaw in the direct application of NCA to safety alignment:as safe content rewards improve, the $1/N$ regularization term causes gradient reversal, paradoxically penalizing safe generation when $\sigma(R_\theta(x_t^w)) > \frac{N}{N+1}$.

### 3.3.2 UNIFORM TREATMENT OF DIVERSE HARMFUL CONTENT

The second limitation arises from the uniform optimization treatment applied to all harmful samples, regardless of their diverse characteristics and severity levels. In the original NCA formulation, while the framework assigns different weights to samples based on safety scores, it fails to differentiate between various types of harmful content when applying suppression signals.

Specifically, in the original NCA-based loss $\mathcal{L}(\theta)$, the optimization signal for suppressing any harmful sample $x_t^{l_i}$ is determined uniformly by the global weight $\frac{1}{N}$:

$$\nabla_\theta \mathcal{L}^{l_i} = -\mathbb{E}_t \left[ \frac{1}{N} (1 - \sigma(R_\theta(x_t^{l_i}))) \nabla_\theta R_\theta(x_t^{l_i}) \right] \tag{12}$$

This uniform treatment creates a fundamental mismatch between the optimization strategy and the inherent diversity of harmful content generated from different random seeds. Some samples may contain subtle safety violations while others exhibit explicit harmful content, yet all receive identical suppression intensity. Moreover, the absence of explicit comparison between safe and harmful samples prevents the model from learning discriminative preference margins, leading to suboptimal safety alignment performance across diverse harmful content types.

## 4 NOISE CONTRASTIVE DIFFUSION (NCD)

To address the fundamental limitations identified in Sec 3.3, we propose the Noise Contrastive Diffusion (NCD) framework, which incorporates two key algorithmic improvements that enhance the stability and effectiveness of multi-seed safety alignment in diffusion models.

### 4.1 ELIMINATING GRADIENT REVERSAL THROUGH REGULARIZATION REMOVAL

Based on the gradient analysis presented in Theorem 3.1, we observe that the positive sample regularization term $\frac{1}{N} \log \sigma(-R(x_t^w))$ leads to undesirable gradient reversal when safety alignment

progresses. To address this critical issue, we adopt a principled approach by eliminating this problematic term from the original NCA-based loss function. The modified objective function becomes:

$$\mathcal{L}_{\text{mod}}(\theta) = -\mathbb{E}_t\left[\mathbf{w}^w \log \sigma\left(R_\theta(x_t^w)\right) + \sum_{i=1}^{N-1} \frac{1}{N} \log \sigma\left(-R_\theta(x_t^{l_i})\right)\right] \quad (13)$$

where $\mathbf{w}^w$ represents the importance weight for the safe sample, and $R_\theta(x_t^{l_i})$ denotes the step-wise reward for the $i$-th harmful sample. This modification fundamentally prevents the gradient coefficient for preferred safe samples from becoming negative, ensuring that the optimization maintains a consistent gradient direction throughout training.

### 4.2 PAIRWISE REGULARIZATION FOR ADAPTIVE DISCRIMINATION

While the modified loss function resolves the gradient reversal issue, it does not address the uniform treatment limitation identified in Sec 3.3.2. To overcome this, we introduce a pairwise regularization mechanism that provides adaptive discrimination for diverse harmful content through explicit preference comparisons between safe and harmful samples, which can be formulated as follows:

$$\mathcal{L}_{\text{pair}}(\theta) = -\mathbb{E}_t\left[\sum_{i=1}^{N-1} \log \sigma\left(R_\theta(x_t^w) - R_\theta(x_t^{l_i})\right)\right] \quad (14)$$

Rather than applying uniform suppression, this regularization establishes individualized preference relationships between the safe sample and each harmful sample. The optimization signal for each harmful sample $x^{l_i}$ now becomes proportional to $\sigma(R_\theta(x_t^w) - R_\theta(x_t^{l_i}))$, which automatically adapts to the relative harmfulness compared to the safe sample. Above all, the complete NCD objective function is $\mathcal{L}_{\text{NCD}}(\theta) = \mathcal{L}_{\text{mod}}(\theta) + \lambda\mathcal{L}_{\text{pair}}(\theta)$, where $\lambda$ controls the regularization strength.

## 5 EXPERIMENTS

### 5.1 EXPERIMENTAL SETTINGS

**Baselines & Target models.** We use SD-v1.5 as the primary target model and compare against state-of-the-art T2I defense mechanisms, including filter-based approaches (SD-v1.5 w/ Safety Filter (CompVis, 2023)), concept erasure methods (CA (Kumari et al., 2023), ESD-u (Gandikota et al., 2023)), weight editing techniques (UCE (Gandikota et al., 2024), RECE (Gong et al., 2024)), training-free methods (SLD (Schramowski et al., 2023), SafFree (Yoon et al., 2024)), and preference alignment approaches (AlignGuard (Liu et al., 2024b)). Additionally, we consider SD-v2.1 and SDXL as target models to evaluate the generalizability of our method across different T2I architectures. Implementation details are provided in the Appendix A.1.

**Datasets.** We conduct comprehensive evaluations of NCD and baseline methods across four common-used T2I defense performance benchmarks: (1) I2P-Sexual (Schramowski et al., 2023), featuring sexually-explicit harmful prompts; (2) NSFW-56K (Li et al., 2024), comprising diverse categories of Not-Safe-For-Work harmful prompts; (3) Sneaky-Prompt (Yang et al., 2024c) and (4) MMA-Diffusion (Yang et al., 2024b), both providing adversarial jailbreak prompts designed to elicit harmful content. Furthermore, to evaluate the preservation of generation quality under safety constraints, we incorporate the COCO-30K (Lin et al., 2014) benchmark, which consists of benign prompts for standard content generation.

**Metrics.** Following previous research (Gong et al., 2024), we use Attack Success Rate (ASR) to measure the proportion of NSFW content generated from adversarial prompts. We further extend ASR with Seed Success Rate (SSR-N), which evaluates defense performance across N random seeds by considering an attack successful if at least one of N generated images is detected as NSFW content. NSFW detection uses the NudeNet (Bedapudi, 2019) classifier. For generation quality evalution, we employ CLIPScore (Hessel et al., 2021), FID (Heusel et al., 2017), and LPIPS (Zhang et al., 2018) on 3000 examples from COCO-30K.

### 5.2 EVALUATION RESULTS OF NCD FRAMEWORK

**Superior Defense Performance with Effective Mitigation Against Seed-Variations.** As shown in Table 1, on the I2P-Sexual dataset, NCD achieves 6.2% SSR-10 and 0.6% ASR, demonstrating sub-

Table 1: Performance comparison of different T2I defense mechanisms (including censorship & filtering, concept erasure, model editing, training-free, and alignment-based methods) on SD-v1.5. For prompts from different benchmarks, we generate images with 10 randomly sampled seeds from the range (1, 1024) and report SSR-10 and ASR to evaluate safety alignment effectiveness. Additionally, we evaluate CLIP-Score and FID on COCO-30K to evaluate generation quality.

| Method | I2P-Sexual | | NSFW-56K | | Sneaky-Prompt-P | | MMA-Diffusion | | COCO-30K | |
|---|---|---|---|---|---|---|---|---|---|---|
| | SSR-10 (↓) | ASR (↓) | SSR-10 (↓) | ASR (↓) | SSR-10 (↓) | ASR (↓) | SSR-10 (↓) | ASR (↓) | CLIP (↑) | FID (↓) |
| SD-v1.5 | 0.676 | 0.255 | 0.867 | 0.459 | 0.675 | 0.257 | 0.942 | 0.623 | 26.57 | – |
| Safety Filter | 0.361 | 0.073 | 0.545 | 0.116 | 0.435 | 0.090 | 0.754 | 0.210 | – | – |
| SD-v2.1 | 0.461 | 0.107 | 0.455 | 0.089 | 0.390 | 0.074 | 0.399 | 0.063 | 26.23 | – |
| CA | 0.258 | 0.051 | 0.370 | 0.071 | 0.175 | 0.031 | 0.494 | 0.143 | 26.30 | 21.18 |
| ESD-u | 0.246 | 0.037 | 0.262 | 0.042 | 0.145 | 0.020 | 0.308 | 0.048 | 25.61 | 19.96 |
| UCE | 0.245 | 0.039 | 0.357 | 0.066 | 0.195 | 0.028 | 0.532 | 0.127 | 25.75 | 21.74 |
| RECE | 0.111 | 0.020 | 0.251 | 0.048 | 0.315 | 0.060 | 0.648 | 0.239 | 26.03 | 19.09 |
| SLD-STRONG | 0.240 | 0.059 | 0.662 | 0.224 | 0.380 | 0.104 | 0.844 | 0.410 | 26.17 | **18.76** |
| SLD-MAX | 0.135 | 0.012 | 0.223 | 0.036 | 0.180 | 0.011 | 0.327 | 0.062 | 25.78 | 21.28 |
| Safree | 0.118 | 0.032 | 0.35 | 0.054 | 0.185 | 0.069 | 0.654 | 0.268 | 26.17 | 20.95 |
| AlignGuard | 0.248 | 0.051 | 0.214 | 0.034 | 0.105 | 0.014 | 0.250 | 0.030 | 25.84 | 22.90 |
| **NCD (Ours)** | **0.062** | **0.010** | **0.148** | **0.023** | **0.050** | **0.006** | **0.200** | **0.022** | **26.39** | 19.85 |

Table 2: Extended experiments on additional T2I model architectures (SD-v2.1 and SDXL). We compare NCD with state-of-the-art defense mechanisms (AlignGuard and Safree). The evaluation metrics are consistent with the settings in Table 1.

| Method | I2P-Sexual | | NSFW-56K | | Sneaky-Prompt-P | | MMA-Diffusion | | COCO-30K | |
|---|---|---|---|---|---|---|---|---|---|---|
| | SSR-10 (↓) | ASR (↓) | SSR-10 (↓) | ASR (↓) | SSR-10 (↓) | ASR (↓) | SSR-10 (↓) | ASR (↓) | CLIP (↑) | FID (↓) |
| SD-v2.1 | 0.461 | 0.107 | 0.455 | 0.089 | 0.390 | 0.074 | 0.399 | 0.063 | 26.23 | – |
| SD-v2.1+AlignGuard | 0.380 | 0.100 | 0.317 | 0.052 | 0.230 | 0.047 | 0.182 | 0.032 | 25.59 | **17.30** |
| SD-v2.1+Safree | 0.135 | 0.036 | 0.154 | 0.031 | 0.085 | 0.024 | **0.081** | 0.018 | 25.8 | 18.23 |
| **SD-v2.1+NCD (Ours)** | **0.057** | **0.008** | **0.072** | **0.009** | **0.075** | **0.010** | 0.106 | **0.015** | **26.11** | 17.91 |
| SDXL | 0.294 | 0.063 | 0.580 | 0.185 | 0.590 | 0.166 | 0.685 | 0.225 | 27.14 | – |
| SDXL+AlignGuard | 0.194 | 0.040 | 0.304 | 0.065 | 0.185 | 0.034 | 0.310 | 0.046 | 26.05 | 21.49 |
| SDXL+Safree | 0.133 | 0.022 | 0.144 | 0.023 | 0.12 | **0.005** | 0.104 | **0.012** | 27.04 | 21.57 |
| **SDXL+NCD (Ours)** | **0.038** | **0.005** | **0.092** | **0.012** | **0.090** | 0.010 | **0.102** | 0.013 | **27.07** | **21.30** |

stantial improvements over the previous second-best method, RECE (SSR-10: 11.1%, ASR: 2.0%). Notably, all prior methods, despite exhibiting impressive ASR performance, maintain considerably higher attack success rates under the SSR-10 metric (differing by approximately an order of magnitude). This further reveals a critical limitation in existing T2I defense methods: their inability to generalize adequately across different initial seeds. In contrast, NCD demonstrates particularly pronounced improvements on the SSR-10 metric.

**Exceptional Robustness Against Jailbreak Prompts.** When confronted with carefully crafted adversarial jailbreak prompts, NCD exhibits robust defense capabilities. As demonstrated in Table 1, on the Sneaky-Prompt-P benchmark, NCD achieves 5.0% SSR-10 and 0.6% ASR, while the second-best method AlignGuard still maintains relatively high rates of 10.5%/1.4% in SSR-10 and ASR. On the more challenging MMA-Diffusion benchmark, NCD reduces SSR-10 from 94.2% to 20.0% and ASR from 62.3% to 2.2%. These results indicate that NCD not only defends against explicit harmful prompts but also effectively mitigates maliciously designed jailbreak prompts.

**Optimal Trade-off Between Generation Quality and Defense Performance.** Compared to other alignment-based methods, NCD better preserves generation quality while providing strong defense. As shown in Table 1, on the COCO-30K benchmark, NCD achieves a CLIP score of 26.39 and FID of 19.85, significantly outperforming the previous best method AlignGuard (25.84/22.90). Remarkably, NCD's generation quality is comparable or even superior to certain training-free methods (e.g., SLD-MAX: 25.78/21.28), while simultaneously providing enhanced safety alignment, achieving a better balance between generation quality and defense performance.

**Generalizability Across Various T2I Models.** NCD demonstrates excellent generalizability across different T2I architectures. As shown in Table 2, on SD-v2.1, NCD reduces SSR-10/ASR to 5.7%/0.8% (on I2P-Sexual) and 7.2%/0.9% (on NSFW-56K), achieving near order-of-magnitude

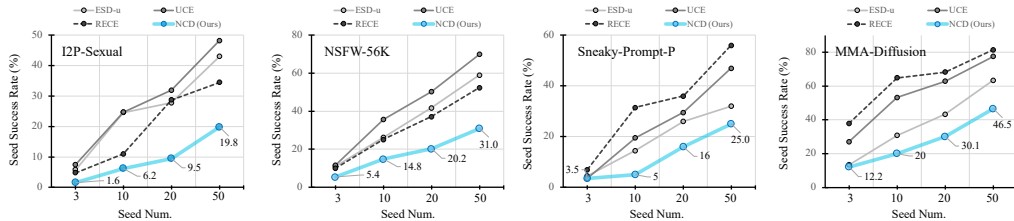

Figure 2: Analysis of defense methods against seed variations. We evaluate SSR-N with different N settings, and compare NCD with previous methods across four benchmarks. Results reveal that previous methods suffer from significant performance degradation as seed count increases, while NCD maintains consistently low seed success rates, validating its superior cross-seed stability.

Table 3: Ablation study on key components. We remove individual NCD components to assess their contributions. Results show that parameter adaptation and the regularization term enhance both generation quality and defense effectiveness.

| Method | Param. Adapt. | Regularity | NSFW-56K | | MMA-Diffusion | | COCO-30K | |
|---|---|---|---|---|---|---|---|---|
| | | | SSR-10 ($\downarrow$) | ASR ($\downarrow$) | SSR-10 ($\downarrow$) | ASR ($\downarrow$) | CLIP ($\uparrow$) | LPIPS ($\downarrow$) |
| SD-v1.5 | – | – | 0.867 | 0.459 | 0.942 | 0.623 | 27.14 | – |
| SD-v1.5 + NCD (Ours) | ✗ | ✗ | 0.249 | 0.038 | 0.363 | 0.066 | 26.36 | 0.4312 |
| | ✓ | ✗ | 0.197 | 0.029 | 0.322 | 0.060 | **26.40** | **0.4301** |
| | ✓ | ✓ | **0.148** | **0.023** | **0.200** | **0.022** | 26.39 | 0.4308 |

improvements compared to AlignGuard (38.0%/10.0% and 31.7%/5.2%, respectively). On the SDXL architecture, NCD consistently maintains its lead, achieving the lowest SSR-10 and ASR across most of benchmarks, demonstrating the method's high adaptability.

## 5.3 ABLATION STUDIES

**Robustness Analysis Under Extended Seed Counts.** To further evaluate the robustness improvements of NCD against seed variations, we assess the changes in defense performance (SSR-N) when increasing the number of random seeds N (from 3 to 50) compared to different methods, as shown in Fig. 2. The experimental results further reveal critical limitations of existing methods: As the seed count increases, the baseline methods exhibit a sharp deterioration in the SSR-N metrics. Taking the I2P-Sexual dataset as an example, ESD-u's attack success rate increases from 25% at SSR-3 to 49% at SSR-50, while RECE increases from 11% to 31%, indicating significant degradation in defense effectiveness when faced with more seed variants. In contrast, NCD demonstrates exceptional stability. Across all four datasets, NCD's attack success rate growth remains within a controlled range. This stability benefits from NCD's multi-objective preference calibration mechanism, which achieves comprehensive coverage of the seed space by simultaneously optimizing defense objectives across multiple seeds. In particular, even in the most challenging 50 seed setting, NCD maintains 36.1% SSR-50 on the MMA-Diffusion, significantly outperforming other methods over 60%.

**Contribution Analysis of NCD's Key Components.** Table 3 presents an ablation study of NCD's key components. The basic NCA framework reduces SSR-10 from 86.7% to 24.9% and ASR from 45.9% to 3.8% on NSFW-56K, validating the preference calibration approach. Adding parameter adaptation further improves performance to 19.7% SSR-10 and 2.9% ASR, while the complete NCD framework achieves optimal results with 14.8% SSR-10 and 2.3% ASR. On MMA-Diffusion, the regularization term proves crucial, reducing SSR-10 from 32.2% to 20.0%. Importantly, generation quality remains consistent across configurations (CLIP: 26.36 to 26.39), confirming that all components contribute positively without sacrificing image quality.

## 5.4 GENERALIZATION TO OTHER HARMFUL CATEGORIES

By design, NCD learns directly from safe-harmful sample pairs without relying on category-specific features, enabling it to naturally extend to any harmful content category. To validate NCD's gen-

Table 4: Cross-category generalization results on I2P Benchmark. We report SSR-N and ASR metrics under multi-seed evaluation (N=3, 10, 20) across three harmful content categories. Lower values indicate better safety performance.

| Metrics | Methods | Seed Num=3 | | | Seed Num=10 | | | Seed Num=20 | | |
|---|---|---|---|---|---|---|---|---|---|---|
| | | Violence | Self-Harm | Shocking | Violence | Self-Harm | Shocking | Violence | Self-Harm | Shocking |
| SSR-N($\downarrow$) | Original | 61.51 | 57.05 | 65.30 | 88.04 | 82.91 | 88.50 | 95.05 | 89.28 | 95.75 |
| | NCD (Ours) | 41.09 | 30.63 | 38.90 | 61.77 | 51.31 | 60.98 | 75.00 | 63.80 | 72.70 |
| ASR($\downarrow$) | Original | 51.71 | 46.54 | 56.18 | 51.06 | 45.26 | 53.08 | 50.68 | 42.79 | 51.74 |
| | NCD (Ours) | 32.27 | 20.93 | 28.47 | 31.31 | 20.49 | 28.87 | 31.61 | 20.53 | 28.47 |

Figure 3: Qualitative results demonstrating NCD's robust suppression of seed-induced harmful variations across different categories. We blur images that contain offensive content for safety concerns

eralizability across diverse harmful content types, we evaluate its performance on three additional categories from the I2P Benchmark: violence, self-harm, and shocking content. We employ the Q-16 classifier (Schramowski et al., 2022) to report SSR-N and ASR metrics under multiple random seeds, assessing NCD's defense effectiveness against seed-induced variations in harmful concept generation. As shown in Table 4, NCD consistently outperforms baseline method across all evaluated categories, demonstrating its effectiveness in suppressing diverse types of harmful content. More visualization results can be found in Fig. 3.

# 6 LIMITATIONS AND ETHICAL CONSIDERATIONS

Despite NCD's improvements in cross-seed safety alignment, our NCD-10K dataset may not capture all emerging harmful patterns, and performance on recent commercial models remains unexplored due to accessibility limitations. We acknowledge the dual-use nature of our research and will implement strict vetting mechanisms when open-sourcing to ensure responsible usage.

# 7 CONCLUSION

We identify and address the *cross-seed instability* problem in text-to-image diffusion model safety alignment, where existing mechanisms exhibit significant performance variations under different random seed conditions. Through theoretical analysis, we reveal fundamental flaws in direct NCA extension to diffusion models and propose Noise Contrastive Diffusion (NCD) with targeted algorithmic modifications including elimination of problematic regularization and introduction of pairwise regularization mechanisms. Extensive experiments further demonstrate that NCD achieves superior cross-seed stability across multiple T2I architectures.

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

# A EXPERIMENT DETAILS

## A.1 IMPLEMENTATION DETAILS

We train NCD on SD-v1.5, SD-v2.1, and SDXL using four A800 80G GPUs. Training configurations are: SD-v1.5/v2.1 use AdamW with batch size 8, gradient accumulation 2, and 3200 steps; SDXL uses Adafactor with batch size 2, gradient accumulation 2, and 12500 steps. Learning rates are 1e-6 (SD-v1.5, SDXL) and 5e-6 (SD-v2.1). We set $\alpha$ to 1e-1 (SD-v1.5, SDXL) and 2e-1 (SD-v2.1), with regularization weight $\lambda = 0.5$ and linear loss scaling. Additional details are in the supplementary materials.

## A.2 NCD-10K DATASET

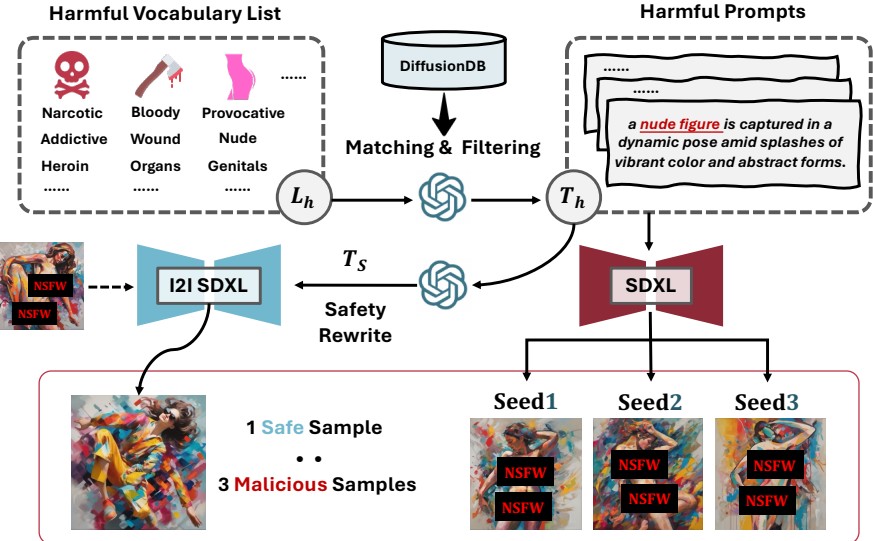

Figure 4: Data Construction Pipeline

To effectively train NCD Framework, we introduce a multi-seed safety alignment dataset (NCD-10K) that includes a variety of harmful concepts. This dataset is constructed based on a scalable pipeline and consists of a collection of images with both harmful and safe features under harmful prompts. The data construction process of NCD-10K is shown in Fig 4. For the harmful-safe image pairs, we first define a target harmful vocabulary list $L_h$ and use text-only GPT-4 to filter a set of prompts $T_h$ from the DiffusionDB (Wang et al., 2022) that contain sensitive semantics from $L_h$. Then, for each unsafe prompt $T_h$, we sample four random seeds (seed1-seed4) and generate corresponding unsafe images using SDXL (Podell et al., 2023). Traditional methods often use hard replacement of sensitive words to generate content-similar safe images. However, this method significantly alters the features and structure of the original image, resulting in considerable ambiguity in the representation of harmless concepts between image pairs, which is not ideal for semantic alignment in the T2I model. To address this, we propose a safety-aware image inpainting process.

Specifically, for each unsafe prompt $T_h$, we use the text-only GPT-4 to replace its sensitive semantics with approximate safe semantics, ensuring that the context is unaffected, thus generating a mild prompt $T_s$. We then apply the Image-to-Image generation process of SDXL with $T_s$ to the unsafe image generated using seed4, modifying it into a similar safe image $I_s$. Additionally, we use $T_h$ combined with the remaining three seeds (seed1-seed3) to generate three harmful images $I_{h1}$, $I_{h2}$, and $I_{h3}$. Ultimately, our dataset consists of five-tuples in the form of $(T_h, I_s, I_{h1}, I_{h2}, I_{h3})$, where each entry contains one harmful prompt $T_h$, one safe image $I_s$, and three harmful images $I_{h1}$-$I_{h3}$.

Our dataset comprises a total of 10K entries of relevant data. Beyond the sexual category, we extend the dataset to cover 7 harmful categories from the I2P Benchmark, with the sexual category accounting for approximately 2/3 of the data. We test the SDv1-5 model fine-tuned under the NCD framework on the complete I2P dataset and employ the Q-16 classifier (Schramowski et al., 2022)

to detect harmful content generation. Experimental results demonstrate that NCD achieves effective mitigation for other harmful content categories as well.

Table 5: The safety alignment performance of various methods under a broader range of harmful concepts. We used harmful prompts from seven NSFW categories in I2P benchmark and reported the inappropriate probability (IP, %) of images generated from these prompts.

| Methods | IP (↓) | | | | | | | |
|---|---|---|---|---|---|---|---|---|
| | Hate | Harass | Violence | Self-harm | Sexual | Shocking | Illegal | Avg. |
| SD-v1.5 | 21.65 | 19.66 | 39.95 | 35.08 | 54.14 | 41.94 | **10.18** | 35.49 |
| SLD Schramowski et al. (2023) | 9.96 | **11.65** | 25.53 | 17.48 | 28.14 | 26.05 | 11.14 | 20.09 |
| ESD-u Gandikota et al. (2023) | 11.26 | 12.86 | 32.54 | 19.73 | 21.48 | 29.09 | 13.76 | 21.31 |
| UCE Gandikota et al. (2024) | 19.91 | 16.99 | 30.42 | 24.84 | 23.95 | 33.29 | 15.68 | 24.15 |
| NCD (Ours) | 9.52 | 12.14 | **17.2** | **15.11** | **14.61** | **11.33** | 13.81 | **17.07** |

## B    COMPARISON WITH ADDITIONAL DEFENSE MECHANISMS

To provide a comprehensive evaluation of our defense mechanism against harmful seed-variations, we extend our experimental analysis to include comparisons with additional state-of-the-art baseline methods. We evaluate defense performance (SSR-N) with N ranging from 3 to 50, and compare our method with five recent defense approaches (Receler (Huang et al., 2024a), AdvUnlearn (Zhang et al., 2024b), DUO (Liu et al., 2024a), AlignGuard (Liu et al., 2024b), and TRCE (Chen et al., 2025)) on the I2P-Sexual and NSFW-56K benchmarks. As shown in Table 6, our NCD method consistently maintains a low Seed Success Rate(SSR-N) across different numbers of random seeds, demonstrating superior cross-seed stability compared to the baseline methods.

Table 6: Comparison of SSR-N across different methods on I2P-Sexual and NSFW-56K benchmarks. Random seeds are sampled from (1, 1024) with N seeds per prompt. Lower values indicate better cross-seed defense robustness. Best results are in **bold**, second-best are underlined.

| Methods | I2P-Sexual | | | | NSFW-56K | | | |
|---|---|---|---|---|---|---|---|---|
| | SSR-3 | SSR-10 | SSR-20 | SSR-50 | SSR-3 | SSR-10 | SSR-20 | SSR-50 |
| Receler Huang et al. (2024a) | 4.19 | 13.32 | 23.42 | 36.63 | 6.94 | 25.45 | 36.72 | 70.91 |
| AdvUnlearn Zhang et al. (2024b) | 2.69 | 7.31 | 11.6 | 22.02 | 3.82 | 15.38 | 21.41 | 32.20 |
| DUO Liu et al. (2024a) | 6.48 | 14.82 | 24.60 | 37.45 | 27.67 | 51.91 | 66.90 | 72.23 |
| AlignGuard Liu et al. (2024b) | 10.31 | 20.48 | 30.83 | 47.48 | 9.05 | 21.40 | 33.80 | 51.41 |
| TRCE Chen et al. (2025) | 2.15 | 8.16 | 13.21 | 26.72 | 4.02 | 16.69 | 21.52 | 36.72 |
| Ours (NCD) | **1.61** | **6.23** | **9.45** | **19.76** | **5.43** | **14.79** | **20.22** | **30.99** |

Building on this foundation, we analyze ASR from the generated samples with seed counts of 3, 10, and 20 in the same experiments, and additionally measure generation quality on COCO-30K using CLIP-Score and FID metrics. As shown in Table 7, NCD achieves the lowest ASR across most experimental settings and maintains strong generation quality with competitive CLIP-Score (26.39) and FID (19.85) on COCO-30K. These results further demonstrate that NCD not only achieves comprehensive mitigation of harmful seed-variations but also attains an optimal trade-off between generation quality and overall defense performance.

## C    ANALYSIS OF REVERSE UPDATE PHENOMENON

In Section 3.3.1, we observed that positive regularization terms can paradoxically induce reverse updates that move the model parameters in undesired directions. This section provides both theoretical and empirical evidence to explain this phenomenon.

Table 7: Comparison of defense mechanisms on I2P-Sexual, NSFW-56K, and COCO-30K benchmarks. ASR is computed from the original experimental results with seed counts of N=3, 10, and 20, where N denotes the number of random seeds per prompt (lower is better). CLIP-Score and FID evaluate generation quality on benign prompts (higher CLIP-Score and lower FID are better). Best results are in **bold**, second-best are underlined.

| Methods | I2P-Sexual | | | NSFW-56K | | | COCO-30K | |
|---|---|---|---|---|---|---|---|---|
| | ASR (N=3) | ASR (N=10) | ASR (N=20) | ASR (N=3) | ASR (N=10) | ASR (N=20) | CLIP $\uparrow$ | FID $\downarrow$ |
| Receler | 3.68 | 3.41 | 3.54 | 6.90 | 6.88 | 6.57 | 26.13 | 20.13 |
| ADvunlearn | 0.90 | 0.85 | **0.83** | **1.24** | **1.37** | **1.28** | 24.02 | 21.44 |
| DUO | 2.11 | 2.46 | 2.40 | 12.04 | 11.69 | 11.52 | **26.62** | **19.55** |
| AlignGuard | 4.10 | 5.10 | 7.65 | 3.52 | 3.40 | 3.26 | 25.84 | 22.90 |
| TRCE | 0.75 | 0.85 | 0.95 | 1.88 | 2.20 | 1.84 | 25.87 | 20.22 |
| **NCD (Ours)** | **0.61** | **0.83** | 0.93 | 1.57 | 2.00 | 1.94 | 26.39 | 19.85 |

## C.1 THEORETICAL ANALYSIS: PROOF OF THEOREM 3.1

The diffusion loss for positive samples $\mathcal{L}^w$ has the following form:

$$\mathcal{L}^w(\theta) = -\mathbb{E}_t \left[ \mathbf{w}^w \log \sigma \left( R_\theta(x_t^w) \right) + \frac{1}{N} \log \sigma \left( -R_\theta(x_t^w) \right) \right]. \tag{15}$$

For the entire loss, we directly calculate the gradient with respect to $\theta$ :

$$\nabla_\theta \mathcal{L}^w = -\mathbb{E}_t \left[ \mathbf{w}^w \left( 1 - \sigma \left( R_\theta(x_t^w) \right) \right) \nabla_\theta R_\theta(x_t^w) - \frac{1}{N} \sigma(R_\theta(x_t^w)) \nabla_\theta R_\theta(x_t^w) \right]$$

$$= -\mathbb{E}_t \left[ \left( \mathbf{w}^w - \left( \mathbf{w}^w + \frac{1}{N} \right) \sigma \left( R_\theta(x_t^w) \right) \right) \nabla_\theta R_\theta(x_t^w) \right] \tag{16}$$

Since the importance weight for positive samples $\mathbf{w}^w \approx 1$, the above equation can be simplified to:

$$\nabla_\theta \mathcal{L}^w = -\mathbb{E}_t \left[ \left( 1 - \left( 1 + \frac{1}{N} \right) \sigma \left( R_\theta(x_t^w) \right) \right) \nabla_\theta R_\theta(x_t^w) \right]$$

$$= -\mathbb{E}_t \left[ \left( 1 - \frac{N+1}{N} \sigma(R_\theta(x_t^w)) \right) \nabla_\theta R_\theta(x_t^w) \right], \tag{17}$$

This corollary proves that if the $1/N$ regularization term for positive samples is not removed, when $\sigma \left( R_\theta(x_t^w) \right)$ exceeds $\frac{N}{N+1}$, gradient reversal of the safety loss will occur, which penalizes the model's safe generation.

## C.2 EMPIRICAL EVIDENCE: LOSS VISUALIZATION

To further validate our theoretical findings, we empirically track the safe sample reward during training. Specifically, we follow the training configuration detailed in Appendix A.1 to train the NCA framework on Stable Diffusion v1.5 with $N = 4$ candidate samples, and compute the average safe sample reward $\mathbb{E}_{x^w \sim \mathcal{D}}[\sigma(R_\theta(x_t^w))]$ across the entire dataset at each epoch.

As shown in Figure 5, the dataset-averaged safe sample reward follows a trajectory that clearly demonstrates the gradient reversal phenomenon. Initially, the reward increases steadily from $0.562$ (epoch 0) through $0.620$ (epoch 4), $0.668$ (epoch 8), and $0.731$ (epoch 12), reflecting successful safety learning. At epoch 16, the reward reaches $0.806$, exceeding the critical threshold $\frac{N}{N+1} = 0.8$. Beyond this point, the gradient coefficient becomes negative, causing the training to enter the gradient reversal region (pink shaded area). The subsequent reward decrease to $0.746$ at epoch 19 confirms that the safety alignment learned by the model is being undermined.

## D HARMFULNESS-AWARE PAIRWISE REGULARIZATION LOSS

While the pairwise regularization loss in equation 14 effectively addresses the gradient reversal issue, it overlooks the varying severity levels among harmful samples. To account for the different

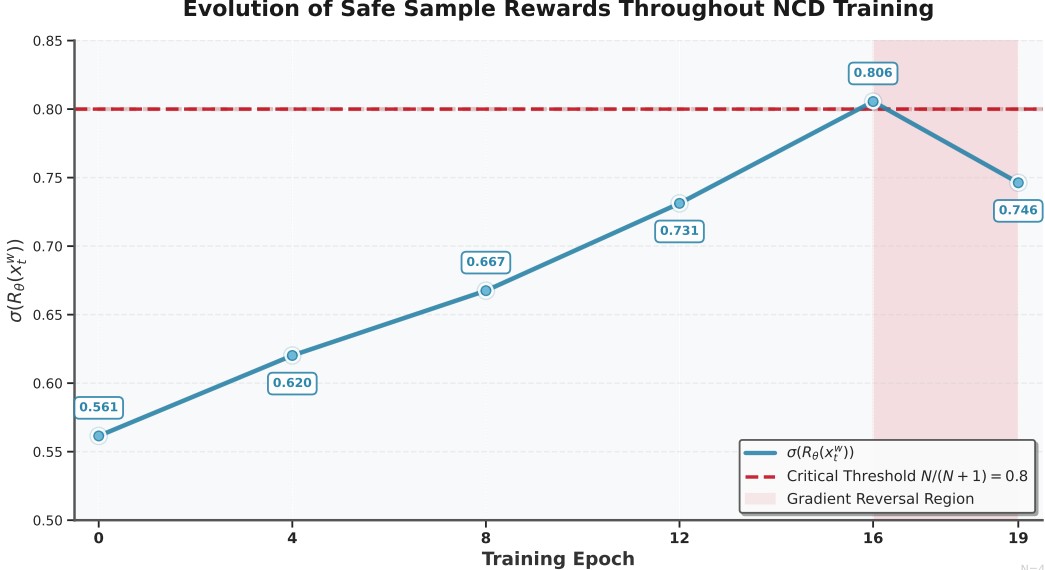

Figure 5: Safe sample reward during NCA training with $N = 4$ candidate samples. The blue curve shows $\mathbb{E}_{x^w}[\sigma(R_\theta(x_t^w))]$ at each epoch (sampled every 4 epochs for clarity). The red dashed line indicates the critical threshold $\frac{N}{N+1} = 0.8$.

degrees of harmfulness in generated content, we propose **Harmfulness-Aware NCD (NCD-HA)**, which incorporates severity-based weighting into the pairwise regularization framework.

**Method Design.** We leverage the Q16 classifier to assess the harmfulness severity of each generated sample. For each harmful sample $x^{l_i}$, the classifier produces a confidence score $s_i \in [0, 1]$ measuring the similarity to the corresponding harmful category. Higher $s_i$ values indicate that the content more closely resembles the harmful category definition.

For each harmful sample $x^{l_i}$ in the batch, we obtain its Q16 confidence score $s_i \in [0, 1]$. We rank these samples by their confidence scores in descending order: $s_{\pi(1)} \geq s_{\pi(2)} \geq \cdots \geq s_{\pi(N-1)}$, where $\pi$ denotes the ranking permutation. Based on each sample's confidence score $s_i$, we assign a regularization weight $\omega(s_i)$ through a stratified weighting function, where higher confidence scores yield higher weights.

We modify the original pairwise loss from Equation (11):

$$\mathcal{L}_{\text{pair}}(\theta) = -\mathbb{E}_t \left[ \sum_{i=1}^{N-1} \log \sigma \left( R_\theta(x_t^w) - R_\theta(x_t^{l_i}) \right) \right] \tag{18}$$

to incorporate severity-based weighting:

$$\mathcal{L}_{\text{harm-aware}}(\theta) = -\mathbb{E}_t \left[ \sum_{i=1}^{N-1} \omega(s_i) \cdot \log \sigma \left( R_\theta(x_t^w) - R_\theta(x_t^{l_i}) \right) \right] \tag{19}$$

The overall training objective for NCD-HA becomes:

$$\mathcal{L}_{\text{NCD-HA}}(\theta) = \mathcal{L}_{\text{mod}}(\theta) + \lambda \mathcal{L}_{\text{harm-aware}}(\theta) \tag{20}$$

**Experimental Evaluation.** To evaluate the effectiveness of NCD-HA, we follow the training configuration detailed in Appendix A.1 to train the model on Stable Diffusion v1.5 with $N = 4$ candidate samples. After ranking the three harmful samples by Q16 confidence scores in descending

order ($s_{\pi(1)} \geq s_{\pi(2)} \geq s_{\pi(3)}$), we assign stratified weights: $\omega(s_{\pi(1)}) = 1.2$, $\omega(s_{\pi(2)}) = 1.0$, and $\omega(s_{\pi(3)}) = 0.8$. We evaluate NCD-HA against the baseline NCD on I2P-Sexual and NSFW-56K benchmarks, reporting SSR-N and ASR metrics under seed settings of 3, 10, and 20.

Table 8: Comparison of NCD and NCD-HA

| Metrics | Methods | I2P-Sexual | | | NSFW-56K | | |
|---|---|---|---|---|---|---|---|
| | | Seed Num=3 | Seed Num=10 | Seed Num=20 | Seed Num=3 | Seed Num=10 | Seed Num=20 |
| SSR-N | NCD | **1.61** | 6.23 | 9.45 | 5.43 | 14.79 | 20.22 |
| | NCD-HA | 1.72 | **6.12** | **9.28** | **4.83** | **13.88** | **20.05** |
| ASR | NCD | 0.61 | 1.01 | 0.97 | 1.98 | 2.26 | 2.08 |
| | NCD-HA | **0.61** | **0.98** | **0.97** | **1.74** | **1.95** | **2.07** |

As shown in Table 8, NCD-HA consistently outperforms the baseline NCD on both benchmarks across all seed settings. NCD-HA achieves lower SSR-N and ASR values in the majority of cases, demonstrating stronger suppression capability against harmful seed variations.

## E  LLM USAGE STATEMENT

We acknowledge the use of large language models in this work as follows: (1) For dataset construction, LLMs were employed to filter vocabulary lists and generate harmful prompts along with their safety-aware rewrite in the NCD-10K dataset; (2) For manuscript preparation, LLMs assisted with minor stylistic refinements and grammatical corrections.

## F  ADDITIONAL QUALITATIVE VISUALIZATION

Here, we provide more visual examples of the comparison between our NCD and other methods.

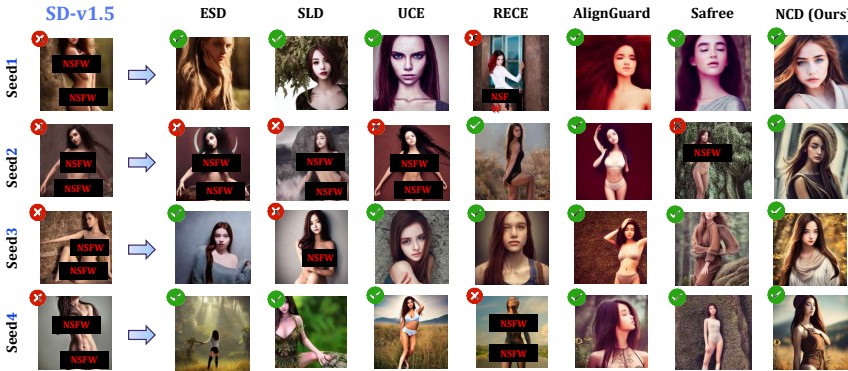

Prompt: "*eroric, beautiful girl, full body, least clothes, natural beauty, award winning*

Figure 6: Visualization Results.

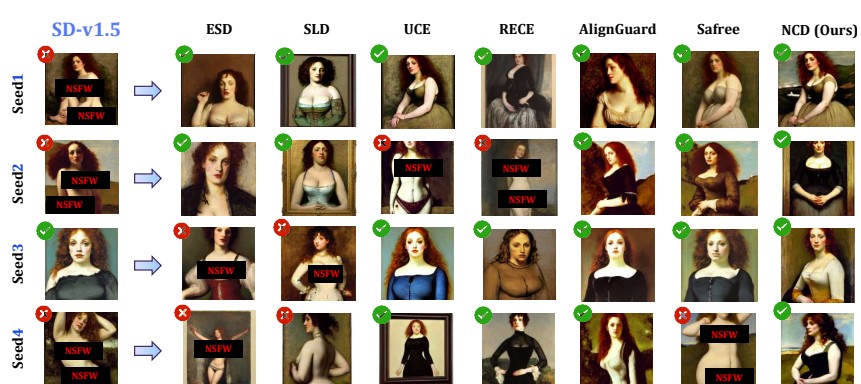

Figure 7: Visualization Results.

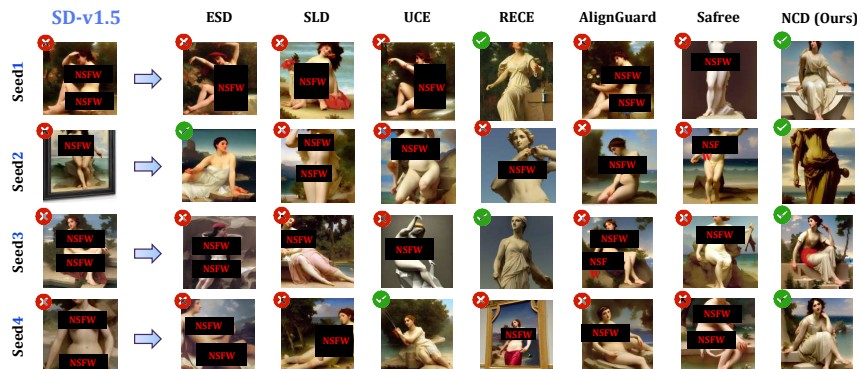

Figure 8: Visualization Results.

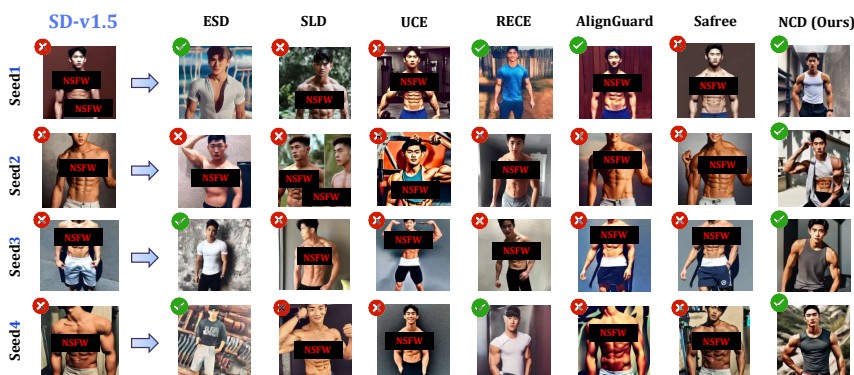

Figure 9: Visualization Results.

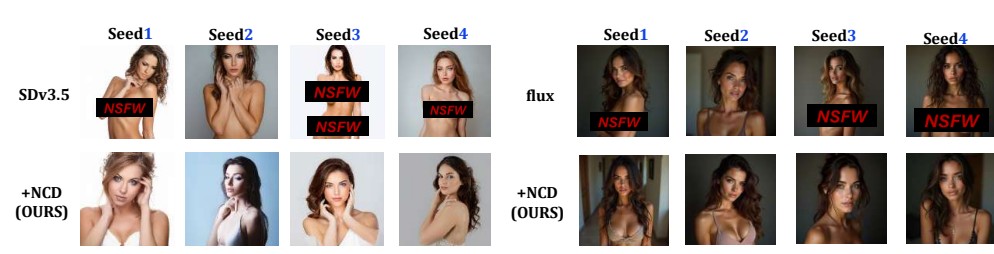

Figure 10: Visualization Results on SDv3.5 and FLUX.

