# OpenReview forum: "Towards Seed-Invariant Safety Alignment in Text-to-Image Models"
_ICLR.cc/2026/Conference — ICLR 2026 Conference Withdrawn Submission_

### Official Review · Reviewer_cxrR · 2025-10-28

**Soundness:** 3
**Presentation:** 3
**Contribution:** 3
**Rating:** 6
**Confidence:** 4

**Summary:**

This paper focuses on a critical yet overlooked issue in text-to-image (T2I) diffusion models: cross-seed safety alignment instability. Specifically, the same malicious prompt may generate diverse harmful image variants under different random seeds, while existing safety mechanisms (e.g., filters, concept erasure, model editing) exhibit significant variations in defense effectiveness against these variants, creating vulnerabilities in real-world deployment.

The authors first attempt to extend the recently proposed Noise Contrastive Alignment (NCA) framework to diffusion models to simultaneously handle multiple harmful samples generated from different seeds. However, theoretical analysis reveals two fundamental flaws in direct extension: First, the gradient reversal problem—the positive sample regularization term in NCA paradoxically penalizes safe content generation as safety alignment progresses. Second, uniform suppression of harmful samples—ignoring severity variations among different harmful samples prevents the model from developing fine-grained discriminative capabilities.

To address these issues, the authors propose Noise Contrastive Diffusion (NCD), which incorporates two core improvements: removing the regularization term causing gradient reversal, and introducing a pairwise regularization mechanism that establishes individualized preference relationships between safe samples and each harmful variant.

Experiments demonstrate that NCD significantly outperforms existing SOTA methods across multiple benchmarks (I2P-Sexual, NSFW-56K, Sneaky-Prompt, MMA-Diffusion), reducing SSR-10 (the success rate of generating at least one harmful image across 10 seeds) from 11.1% to 6.2%, while maintaining or even surpassing baseline generation quality on COCO-30K (CLIP Score 26.39, FID 19.85). NCD also exhibits strong generalizability on both SD-v2.1 and SDXL.

**Strengths:**

1. This paper is the first to systematically identify and formalize the problem of "seed-invariant safety alignment," revealing the vulnerability of existing methods in this dimension. While NCA has been used in language models, the authors not only identify its theoretical flaws in diffusion models (gradient reversal, uniform suppression) but also propose targeted algorithmic corrections (removing regularization terms + pairwise preferences), representing a creative adaptation and improvement of existing ideas to solve new problems. The problem definition itself is novel and crucial for practical deployment safety.

2. The theoretical analysis is rigorous (e.g., gradient derivation in Theorem 3.1), and the experimental design is comprehensive: it covers various types of safety mechanisms (filtering, concept erasure, editing, alignment); uses multiple authoritative harmful prompt benchmarks, including adversarial jailbreak datasets; introduces SSR-N (Seed Success Rate), an evaluation metric closer to real-world risks; validates generalizability and robustness across models (SD-v1.5/v2.1/XL) and seed counts (N=3–50); provides ablation studies to verify the contribution of each component; and constructs the NCD-10K dataset, which includes safe/harmful image pairs and multi-seed variants with a reasonable pipeline (using GPT-4 for safe rewriting + image inpainting).

3. As T2I models are widely deployed, their safety robustness is critical. The "cross-seed instability" revealed in this paper is a significant blind spot in existing safety solutions, and NCD provides an effective and practical resolution.

**Weaknesses:**

The coverage of the NCD-10K dataset is limited: although it includes seven categories of harmful content, sexual content accounts for two-thirds (page 13), and the dataset is filtered based on DiffusionDB and GPT-4. This may lead to an overestimation of the model's generalization capability on other categories (e.g., hate, illegal). Table 4 shows that NCD achieves limited improvement in categories such as "Self-harm" and "Hate" (e.g., Self-harm improved from 39.95% → 17.2%), but the reasons are not thoroughly analyzed. It is recommended to include specialized evaluations for long-tail harmful categories.

**Questions:**

Can you provide dynamic curves showing the safe sample reward Rθ(x^w_t) and its corresponding gradient direction throughout the training process? This evidence would directly demonstrate the practical impact described in Theorem 3.1.

While NCD eliminates the uniform negative sample regularization term from NCA, does the pairwise loss (Eq. 11) inherently treat all harmful samples equally? For potentially more dangerous harmful samples, could a severity-based weighted pairwise loss be incorporated?

Figure 2 indicates that NCD still maintains a 36.1% failure rate on MMA-Diffusion under SSR-50 conditions. Does a theoretical upper limit exist for this performance? Could further improvements be achieved by increasing the number of training seeds or implementing adversarial seed sampling?

Have comparisons with multi-response extensions of DPO (such as IPO and SimPO) been considered? These methods might also help address seed instability but weren't included in the experimental comparisons.

While the paper mentions plans for open-source release with strict review protocols, would there be commitment to publishing a safe subset of NCD-10K that excludes genuinely harmful images to facilitate community research?

---

### Official Review · Reviewer_dHZV · 2025-10-31

**Soundness:** 3
**Presentation:** 3
**Contribution:** 3
**Rating:** 2
**Confidence:** 5

**Summary:**

In this paper, the authors observed an interesting phenomenon that erased models can generate harmful images under different random seeds for harmful inputs. Then they introduce Noise Contrastive Alignment and analyze its drawbacks mathmatically, and further propose Noise Contrastive Diffusion to address these problems.

**Strengths:**

1. The paper is well-written, easy for readers to follow.
2. The paper introduce an interesting phenomenon.
3. The paper propose some incremental optimizations on the DPO and NCA algorthm.

**Weaknesses:**

1. Relationship between the phenomenon and the proposed method. I acknowledge that the cross-seed instability phenomenon has not been intruduced in previous studies, but I think the authors did not provide a targeted explanation. They only said that "current approaches fail to establish *robust* safety alignment" but it has also been mentioned many times in previous studies. In fact, in the training, we usually do not set random seed in each training iteration. Therefore, the seed is changing also in the training. In other words, I want to know what is the difference between the reason for this observation and for the robustness issue?

2. Introduction of pair-wise dataset. NCA is proposed to address  the limitation of DPO methods which can only handle pairwise preference data but this paper propose NCD to introduce pairwise data again. It increase the difficulty of implementing this method, as evidenced in Appendix A.2. And the authors themselves also said that " our NCD-10K dataset may not capture all emerging harmful patterns".

3. The baselines are old. Please compare your methods with more recent methods, especially adversarial methods such as Receler and AdvUnlearn.

**Questions:**

See weaknesses. If my concerns can be address, I will change my rating.

---

### Official Review · Reviewer_AdHF · 2025-10-31

**Soundness:** 2
**Presentation:** 2
**Contribution:** 2
**Rating:** 4
**Confidence:** 5

**Summary:**

This paper is the first to discuss the problem of insufficient robustness in current T2I safety mechanisms when facing the influence of random seeds, and it attempts to solve this using preference alignment. Compared to naive DPO, this method proposes the Noise Contrastive Diffusion (NCD) Framework, to achieve a) eliminating the problematic regularization term to prevent gradient reversal and b) introducing a pairwise regularization mechanism to establish individualized preference relationships between the safe sample and each harmful variant.

**Strengths:**

1. The paper addresses a practical problem: cross-seed instability in T2I safety alignment.
2. The method is well-motivated, with a theoretical analysis (Theorem 3.1) of the flaws in a direct NCA extension (i.e., gradient reversal).
3. The proposed Seed Success Rate (SSR-N) metric is a reasonable and valuable tool for evaluating this specific problem.
4. The experimental results are comprehensive and demonstrate state-of-the-art performance across multiple models.

**Weaknesses:**

1. Although the authors propose a seemingly practical task setting, the argument that "existing safety alignment methods are vulnerable when generating with different seeds" is not well-supported: a) The authors only provide a few generated examples and lack an analysis of _why_ existing methods are vulnerable in this scenario. b) Is this problem equivalent to the problem of "insufficient safety in existing alignment methods"? In fact, the authors only show a few early concept erasure methods and overlook the discussion of methods with stronger safety capabilities[1,2,3,4]. With more powerful methods, the risk bring by seed may be small.
2. The practical novelty of NCD is questionable. The final forms of Eq10 and Eq11 appear to be a variation of (1, N) Direct Preference Optimization (DPO). This paper fails to clearly differentiate its contribution from this baseline.
3. Considering Weakness 1, the authors have actually overlooked a baseline directly related to the task (DUO[5]). This work also improves DPO techniques for T2I safety alignment tasks.
4. The experimental evaluation has some confusing points. First, in Figure 3, the training data shown includes NSFW concepts other than sexual content (e.g., drug, bloody, etc.). The test datasets used are also multi-concept NSFW datasets. All indications suggest the authors are trying to demonstrate that the proposed method is compatible with aligning various types of NSFW concepts. However, when conducting quantitative evaluation and visualization, they only consider the sexual concept (evaluated by NudeNet).
5. Following up on Weakness 4, the authors should consider adding evaluations for other NSFW concepts. Additionally, there should be some discussion on the method's compatibility with important tasks like celebrity and style.
6. Newer models such as SD3.5 and FLUX should also be included in the discussion.

[1] Defensive Unlearning with Adversarial Training for Robust Concept Erasure in Diffusion Models, nips24
[2] Localized Concept Erasure for Text-to-Image Diffusion Models Using Training-Free Gated Low-Rank Adaptation, cvpr25
[3]  TRCE: Towards Reliable Malicious Concept Erasure in Text-to-Image Diffusion Models, iccv25
[4] Safetydpo: Scalable safety alignment for text-to-image generation, iccv25
[5] Direct Unlearning Optimization for Robust and Safe Text-to-Image Models, nips24

**Questions:**

Please address the weaknesses.

---

### Official Review · Reviewer_DEWM · 2025-11-03

**Soundness:** 3
**Presentation:** 3
**Contribution:** 2
**Rating:** 4
**Confidence:** 4

**Summary:**

This paper focuses on the safety challenges faced by text-to-image diffusion models due to cross-seed instability, where malicious prompts generate diverse harmful variants.  It reveals flaws in extending Noise Contrastive Alignment (NCA) to diffusion models, including gradient reversal from positive regularisation and uniform suppression of harmful samples. Therefore, a method called Noise Contrastive Diffusion (NCD) is proposed to address issues by eliminating problematic regularisation and introducing pairwise regularisation mechanisms.

**Strengths:**

This work aims to bridge the gap in applying the NCA method, which was originally designed for language models, to diffusion models. By analysing the back-propagation in NCA, they identify a "Gradient Reversal" issue in the original framework and propose a modified version for diffusion models to avoid such an issue. Furthermore, a "Uniform Treatment" issue is also investigated, which is then addressed by a pairwise regularisation.

**Weaknesses:**

1. Firstly, the proposed methods cannot guarantee the invariance of the models or explicitly optimise them for invariance. The primary challenges this work aims to address are the cross-seed instability of existing safety mechanisms. However, the proposed method did not explicitly focus on the instability or invariance. Instead, it merely included more negative samples by extending the NCA method, which appears to be a "brute-force" approach. Furthermore, the core idea of NCA is to optimise the absolute likelihood for each response rather than adjusting the relative likelihood across different responses, which does not inherently provide a guarantee of invariance. The paper conducts a theoretical analysis of back-propagation in NCA, but it focuses on the gradient direction and does not explicitly address the crucial question: how to guarantee stability and invariance.

2. Secondly, the experiment results further demonstrate that the proposed method did not effectively address the instability issue. As shown in Fig. 2, the paper proposed a novel SSR-N metric. While the proposed method consistently achieves lower ASR compared to previous methods, the trend of ASR exhibits a clearly increasing pattern as the number "N" increases, similar to previous methods. There is no clear suppression of this increasing trend or a plateau. This highlights the drawback of simply increasing the number of negative samples, as the number of malicious attempts increases, the defence becomes weakened.

3. Ideally, the derivation of Theorem 3.1 should be provided. Although this derivation is relatively straightforward, providing a detailed process would facilitate readers in verifying its correctness.

**Questions:**

Please address the weaknesses.

---

### Author Response · Authors · 2025-12-03

***Dear Area Chair and Reviewers,***

We sincerely thank the **Area Chair**, and **all reviewers** for their constructive feedback and the time invested in reviewing our paper.
To ensure we address every reviewer comment as comprehensively as possible and facilitate a streamlined assessment for the Area Chair to minimize the extra workload caused by technical issues, we have consolidated our major revisions and detailed point-by-point responses into this single communication. We hope this organized format allows for an efficient review of our extensive rebuttal efforts.

Below, we highlight the pioneering nature of our contributions and detail the comprehensive revisions made to address the reviewers' concerns. **(Detailed point-by-point responses to each reviewer are appended immediately following this summary.)**

---

***Summary of Contributions***

Our work aims to bridge a critical gap in T2I safety by addressing the "cross-seed instability" problem. Our core contributions are:

- **Pioneering Identification of Cross-Seed Instability:** We are the first to systematically identify and formalize the cross-seed instability phenomenon, revealing a fundamental vulnerability where existing safety mechanisms fail to defend against diverse harmful variants generated from different noise initializations.
- **Theoretical Breakthrough in Optimization Stability:** We identify a critical gradient reversal pathology inherent in direct multi-seed alignment extensions. Our theoretical analysis proves that this pathology paradoxically penalizes safe content generation. By eliminating the problematic regularization, NCD theoretically guarantees optimization stability, overcoming a barrier that plagues baseline methods.
- **Robust Seed-Invariant Framework:** We propose Noise Contrastive Diffusion (NCD), which introduces a novel multi-candidate pairwise regularization mechanism. Unlike standard uniform suppression, this mechanism explicitly guides the model to learn a robust discriminative preference margin across multiple harmful variants.
- **New SOTA Performance:** Extensive experiments demonstrate that NCD establishes a new state-of-the-art for stability and achieves the optimal trade-off between rigorous safety and exceptional generation quality.

---

***Summary of Revisions & Comprehensive Responses***

We have made significant efforts to strengthen the manuscript, conducting extensive new experiments and analyses to address the collective concerns of the reviewers:

* **Expanded SOTA Comparisons (Reviewer AdHF, dHZV, cxrR):**
    We added four stronger baselines and conducted comprehensive comparisons against **AdvUnlearn, TRCE, DUO, and AlignGuard**. The results (added to the revised manuscript) confirm that cross-seed vulnerability persists even in these advanced models and that NCD consistently maintains superior stability across all settings.
* **Rigorous Theoretical & Empirical Validation (Reviewer DEWM, cxrR):**
    To solidify our theoretical claims, we added the **detailed mathematical derivation of Theorem 3.1** in Appendix C.1. Furthermore, we implemented dynamic visualizations of the safe sample reward trajectory (Figure C.1), which empirically validates the gradient reversal phenomenon and the stability of our solution.
* **Generalization to Long-Tail Categories (Reviewer cxrR):**
    To demonstrate robustness beyond sexual content, we added **Sect 5.4** and **Appendix Table 5**, providing quantitative metrics and visualizations for **Violence, Self-harm, and Shocking** categories. The results prove NCD’s effective generalization to diverse harmful concepts.
* **Methodological Depth & Extensions (Reviewer cxrR):**
    We explored severity-based weighting by implementing and evaluating **Harmfulness-Aware NCD (NCD-HA)** in Appendix D. While adaptive weighting offers valid perspectives, our experiments confirmed that the original NCD design remains a highly rational and robust baseline.
* **Clarifications on Novelty & Mechanism (Reviewer DEWM, AdHF):**
    We clarified the distinction between our NCD and standard DPO, emphasizing NCD's unique capability to capture harmful seed distributions. We also elucidated the fundamental difference between "cross-seed instability" and general safety robustness.

---

We hope these extensive revisions and additional experiments satisfactorily address the reviewers' concerns and clearly demonstrate the value of our work.

Sincerely,

The Authors

---

> ### Author Response · Authors · 2025-12-03
> **Response to Reviewer DEWM**
>
> ***To Reviewer DEWM:***
>
> ***W1: The method lacks an explicit optimization or theoretical guarantee for cross-seed invariance, relying instead on a "brute-force" extension of negative samples.***
>
>   We respectfully clarify that NCD is not simply a "brute-force" extension with more negative samples. One core contribution of NCD lies in identifying and resolving the *gradient reversal* problem that fundamentally undermines training stability, which directly addresses cross-seed invariance at the optimization level.
>
>   Specifically, our theoretical analysis (Theorem 3.1) reveals that when $\sigma(R_\theta(x_t^w)) > \frac{N}{N+1}$, the gradient coefficient becomes negative, causing the model to *penalize* safe content generation. By eliminating the problematic regularization term, NCD ensures consistent gradient directions throughout training, which is the prerequisite for stable multi-seed alignment.
>
>   To empirically validate this, we tracked the safe sample reward $E_{x^w}[\sigma(R_\theta(x^w_t))]$ during NCA training with $N=4$:
>
> | Epoch | 0 | 4 | 8 | 12 | 16 | 19 |
> | :--- | :---: | :---: | :---: | :---: | :---: | :---: |
> | Safe Reward | 0.562 | 0.620 | 0.668 | 0.731 | ***0.806*** | ***0.746*** |
>
> The reward increases steadily until epoch 16, where it reaches 0.806, exceeding the critical threshold $\frac{N}{N+1}=0.8$. Beyond this point, the reward *decreases* to 0.746, confirming that gradient reversal undermines safety alignment. NCD's regularization removal directly prevents this pathological behavior, providing a principled guarantee for training stability rather than relying on brute-force sample scaling. The complete visualization results are provided in Appendix C.2 of the revised manuscript.
>
> ---
>   ***W2: Experimental results show ASR still increases with the number of seeds, indicating the method fails to effectively suppress instability trends.***
>
>   We apologize for a mistake in Figure 2, the y-axis should be "SSR-N" (Seed Success Rate), not "ASR" (Attack Success Rate). This has been corrected in the revised manuscript. We further clarify that ASR-N represents the average attack success rate across N seeds, while SSR-N measures the probability that at least one harmful image appears among N seeds. Mathematically, SSR-N necessarily exhibits an increasing trend with respect to N regardless of the defense method, which is an inherent statistical property of the metric. Based on this, comparisons across methods should focus on: (1) whether ASR remains stable as N increases, and (2) relative SSR-N performance across methods. Our experiments show that NCD's ASR consistently remains at low levels (0.6%-1.0% on I2P-Sexual), while baseline methods exhibit higher and less stable ASR. On the other hand, NCD achieves substantially lower SSR-N at every N value, which sufficiently demonstrates that NCD provides the strongest cross-seed stability among all evaluated methods.
>
>   ---
>
>   ***W3: The detailed derivation for Theorem 3.1 is missing and necessary for verification.***
>
>   Thank you for pointing this out. We have included the detailed derivation of Theorem 3.1 in Section 3.3.1 and Appendix C.1 of the revised manuscript. Specifically, we derive the gradient by directly differentiating the safe sample loss component, then simplify under the condition that the importance weight $\omega^{w} \approx 1$, which reveals the critical threshold $\sigma(R_\theta(x_{t}^{w})) > \frac{N}{N+1}$ at which gradient reversal occurs. Please refer to these sections for the complete proof.

---

> ### Author Response · Authors · 2025-12-03
> **Response to Reviewer AdHF (1/2)**
>
> ***To Reviewer AdHF (1/2):***
>
>   ***W1 & W3: The claim of "cross-seed vulnerability" is weakly supported and fails to account for stronger safety baselines (AdvUnlearn, TRCE, AlignGuard, DUO) that may not suffer from this issue.***
>
>  We fully understand your concerns and address them as follows:
>
> - **(a) Cross-seed instability is a distinct problem, not merely "insufficient safety."**
> Cross-seed instability refers to the phenomenon where a single malicious prompt generates diverse outputs across different noise initializations (some safe, some harmful) even when the *average* ASR appears low. This is fundamentally different from overall safety insufficiency. As shown in our experiments, methods like AdvUnlearn achieve low ASR (0.85% at N=10), yet their SSR-10 reaches 7.31%, meaning harmful content still leaks through specific seed combinations. This inconsistency poses practical risks: attackers can simply iterate through seeds to find vulnerable ones.
>
> - **(b) & W3. Comparisons with stronger recent methods confirm the problem persists.** Following your suggestion, we have conducted comprehensive comparisons with recent state-of-the-art methods including AdvUnlearn, TRCE, AlignGuard, and DUO. We confirm that even these powerful methods exhibit significant cross-seed vulnerability, confirming that cross-seed instability is not resolved by simply improving overall safety strength. Crucially, in evaluating the necessary safety-quality trade-off, our NCD delivers the best cross-seed stability (lowest SSR-N) across all settings while avoiding the generation quality sacrifices seen in other high-safety methods (like AdvUnlearn). By maintaining competitive generation quality compared to high-utility methods (like DUO), NCD achieves the optimal balance between robust safety and model utility. This demonstrates that our multi-seed alignment framework effectively addresses this previously overlooked vulnerability with an optimal trade-off solution.
>
>
>
>
>
>
> | Metrics | Methods | I2P-Sexual(N=3) | I2P-Sexual(N=10) | I2P-Sexual(N=20) | NSFW-56K(N=3) | NSFW-56K(N=10) | NSFW-56K(N=20) |
> | :--- | :--- | :---: | :---: | :---: | :---: | :---: | :---: |
> | **SSR-N**  | Receler | 4.19 | 13.32 | 23.42 | 6.94 | 25.45 | 36.72 |
> | | AdvUnlearn | 2.69 | 7.31 | 11.60 | 3.82 | 15.38 | 21.41 |
> | | DUO | 6.48 | 14.82 | 24.60 | 27.67 | 51.91 | 66.90 |
> | | AlignGuard | 10.31 | 20.48 | 30.83 | 9.05 | 21.40 | 33.80 |
> | | TRCE | 2.15 | 8.16 | 13.21 | 4.02 | 16.69 | 21.52 |
> | | **NCD (Ours)** | **1.61** | **6.23** | **9.45** | **5.43** | **14.79** | **20.22** |
>
>
> | Metrics | Methods | I2P-Sexual(N=3) | I2P-Sexual(N=10) | I2P-Sexual(N=20) | NSFW-56K(N=3) | NSFW-56K(N=10) | NSFW-56K(N=20) | CLIP$\uparrow$ | FID$\downarrow$ |
> | :--- | :--- | :---: | :---: | :---: | :---: | :---: | :---: | :---: | :---: |
> | **ASR**  | Receler | 3.68 | 3.41 | 3.54 | 6.90 | 6.88 | 6.57 | 26.13 | 20.13 |
> | | AdvUnlearn | 0.90 | 0.85 | **0.83** | **1.24** | **1.37** | **1.28** | 24.02 | 21.44 |
> | | DUO | 2.11 | 2.46 | 2.40 | 12.04 | 11.69 | 11.52 | **26.62** | **19.55** |
> | | AlignGuard | 4.10 | 5.10 | 7.65 | 3.52 | 3.40 | 3.26 | 25.84 | 22.90 |
> | | TRCE | 0.75 | 0.85 | 0.95 | 1.88 | 2.20 | 1.84 | 25.87 | 20.22 |
> | | **NCD (Ours)** | **0.61** | **0.83** | 0.93 | 1.57 | 2.00 | 1.94 | 26.39 | 19.85 |
>
>
> ---
>
>
>   ***W2: The novelty is questionable as the proposed loss (Eq. 10/11) closely resembles (1, N) Direct Preference Optimization (DPO).***
>
>   We appreciate the reviewer's observation regarding the structural resemblance to Direct Preference Optimization (DPO). While both NCD and DPO are rooted in preference-based alignment, NCD introduces a fundamental divergence from standard DPO's single pairwise comparison by utilizing an $N$-candidate contrastive learning framework (1 safe sample contrasted against $N-1$ harmful seed candidates). This multi-candidate design significantly enhances novelty by allowing the loss to capture the diverse distribution of harmful seed variations, thereby providing a much richer contrastive signal that guides the model toward a more robust safety boundary. Furthermore, our absolute form of the loss function offers a distinct technical advantage over the relative form used in DPO, as it effectively mitigates potential damage to overall generation quality, a common concern when applying DPO's comparative methods. While we acknowledge that generating the multi-candidate data adds initial complexity, this is an offline, one-time procedure that does not impact training speed. We will mitigate this burden on future research by open-sourcing the complete data processing pipeline and the resulting dataset.

---

> ### Author Response · Authors · 2025-12-03
> **Response to Reviewer AdHF (2/2)**
>
> ***To Reviewer AdHF (2/2):***
>
>   ***W4: Evaluations are overly focused on "sexual" content, ignoring other NSFW categories.***
>
> We sincerely appreciate this valuable suggestion. We acknowledge that our initial focus on the most prevalent sexual content may have inadvertently led to less emphasis on other categories. To further demonstrate the comprehensive capabilities of our method, we have expanded our evaluations in the revised manuscript. Specifically, we added **Sec 5.4** to provide quantitative and qualitative results for Violence, Self-harm, and Shocking content. Furthermore, comprehensive assessments for additional categories, including Hate, Harass, and Illegal, are detailed in Table 5 of the Appendix. These results confirm that our method achieves consistent and excellent improvements across diverse NSFW domains.
>
> ---
>
>   ***W5: Discussion on the method's compatibility with other tasks (e.g., celebrity, style) is missing.***
>
> We thank the reviewer for raising this point. It is important to clarify that NCD, like AlignGuard and DUO, is essentially an improvement to DPO-based techniques for T2I safety alignment. We follow the precedent set by these methods in not prioritizing celebrity or style erasure as core evaluation, as these tasks are fundamentally different from our objective. Moreover, our method is focused on achieving seed-invariant alignment, *i.e.*, eliminating broad unsafe concepts (e.g., sexual content, violence, self-harm) that consistently persist across random seeds. This requirement for robust, global modification of safety knowledge is fundamentally distinct from the fine-grained, instance-level removal required for celebrity or style tasks.

---

> ### Author Response · Authors · 2025-12-03
> **Response to Reviewer dHZV**
>
> ***To Reviewer dHZV:***
>
>   ***W1: The distinction between the proposed "cross-seed instability" and general robustness issues (since seeds change during training) is unclear.***
>
>   Thank you for this insightful question. We clarify that cross-seed instability is fundamentally distinct from general robustness issues in the T2I safety alignment context.  The latter concerns whether safety holds when **inputs change** (e.g., jailbreak or adversarial prompts), whereas our work focuses on whether safety holds when **only the random seed changes** for a fixed, identical prompt. These two problems are orthogonal: a model robust to prompt perturbations may still generate harmful content under certain seed conditions for the same malicious prompt.
>
>   Although seeds naturally vary during training, existing methods only reduce average harmfulness across samples rather than ensuring safety under worst-case seed variations. DPO-based methods process only pairwise preferences, lacking explicit coverage of multi-seed distributions. The proposed NCD specifically addresses this by explicitly optimizing across multiple seed variants simultaneously. Our multi-seed contrastive alignment covers diverse harmful manifestations from different noise initializations, while pairwise regularization establishes preference margins between safe content and *each* harmful variant individually. This transforms the optimization objective from average-case performance to comprehensive coverage of the seed space, directly targeting cross-seed instability.
>
> ---
>
>   ***W2: Re-introducing pairwise data increases implementation difficulty and contradicts the motivation of overcoming DPO's limitations.***
>
>
>
>
> Thanks for your suggestion! We must clarify that our use of pairwise regularization is not a regression to DPO's limitations, but rather a fundamental divergence that enables the solution to cross-seed instability. While DPO is restricted to one-to-one comparisons, NCD utilizes an $N$-candidate contrastive learning framework (1 safe sample contrasted against $N-1$ harmful seed candidates). This multi-candidate design is essential because it allows the loss to capture the diverse distribution of harmful seed variations, thereby providing a much richer contrastive signal that guides the model toward a more robust safety boundary. Although we acknowledge that generating the multi-candidate data adds initial complexity, as referenced in Appendix A.2, this is an offline, one-time procedure that does not impact training speed; we will mitigate this burden on future research by open-sourcing the complete data processing pipeline and the resulting dataset.
>
> Finally, regarding the realistic challenge of capturing all emerging harmful patterns, NCD has already demonstrated effective generalization across several long-tail categories, including Violence, Self-harm, and Shocking (detailed in Section 5.4 and Appendix Table 5). However, we must also acknowledge that achieving comprehensive coverage of all potential vulnerabilities is practically impossible for any method. To maintain defense against emerging risks, we are committed to further expanding and updating our dataset regularly, ensuring continuous adaptation.
>
>
>
> ---
>
>   ***W3: Baselines are outdated; comparisons with recent adversarial methods like Receler and AdvUnlearn are missing.***
>
>
> Thank you for this valuable suggestion. To address it, we have conducted additional comprehensive comparisons with recent state-of-the-art adversarial methods, Receler and AdvUnlearn, across the I2P-Sexual and NSFW-56K benchmarks. The full results, encompassing SSR-N, ASR, and generation quality metrics, are available in the ***Reviewer AdHF W1 & W3***. Our findings confirm that NCD consistently outperforms both Receler and AdvUnlearn across all SSR-N settings ($N=3$ to $N=50$), demonstrating superior cross-seed stability. This advantage becomes particularly pronounced at higher $N$, where these baseline methods suffer significant performance degradation. Furthermore, NCD maintains competitive ASR while simultaneously achieving the optimal trade-off with generation quality (CLIP Score and FID). This robust performance across both stringent safety metrics and utility confirms NCD's superiority over recent adversarial-based methods.

---

> ### Author Response · Authors · 2025-12-03
> **Response to Reviewer cxrR (1/2)**
>
> ***To Reviewer cxrR (1/2):***
>
> ***W1: The NCD-10K dataset is skewed towards sexual content (2/3), potentially overestimating generalization on long-tail categories like "Self-harm".***
>
> We fully understand your concern. Our rationale for focusing mainly on sexual content is that (1) it represents the most prevalent and critical safety risk in diffusion models, and (2) providing a rich dataset in this domain could serve as a helpful resource for community fine-tuning efforts. Moreover, despite the dataset skew, our experiments demonstrate that the data volume reserved for long-tail categories, such as "Self-harm," is sufficient to drive effective unlearning across these concepts. To validate this, we have added Sec 5.4 in the revised manuscript, which provides both quantitative metrics and qualitative visualizations specifically on Violence, Self-harm, and Shocking content. The comprehensive results for even more categories are also detailed in Table 5 of *Appendix*.
>
> ---
>
>   ***Q1: Can you provide dynamic curves of safe sample rewards and gradient directions to validate Theorem 3.1?***
>
> We sincerely appreciate this suggestion! To comprehensively address your concern, we have included the dynamic visualization of safe sample rewards in **Appendix C** (Sec C.2 and **Figure C.1**) of the revised manuscript. These curves empirically track the reward trajectory during training, clearly demonstrating that the gradient reversal phenomenon is triggered once the reward exceeds the critical threshold (e.g., $0.8$ for $N=4$), leading to a subsequent decline in safety alignment. In addition to these visualizations, we have further solidified our claims by providing a detailed **theoretical derivation** (Section C.1), which mathematically substantiates the conditions for this reversal, thus offering a rigorous dual-validation of Theorem 3.1 from both empirical and theoretical perspectives.
>
> ---
>
>   ***Q2: Does the pairwise loss treat all harmful samples equally, and could severity-based weighting be incorporated?***
>
> We sincerely appreciate this valuable suggestion! To verify it, we have incorporated a Harmfulness-Aware Pairwise Regularization Loss (NCD-HA) in the revised manuscript (Appendix D). This method utilizes Q16 classifier confidence scores to assign higher weights to more severe harmful samples, aligning with your recommendation. Experiments demonstrate that NCD-HA provides consistent yet incremental improvements,  effectively lowering ASR on the IP2-Sexual and NSFW-56K dataset. Additionally, we observe that the performance advantage of NCD-HA tends to diminish as the number of seeds ($N$) increases (e.g., at $N=20$), where the original NCD achieves nearly identical results. This finding confirms that while incorporating severity awareness offers a valid optimization perspective, our original NCD may also be a rational design.
>
>
>
>
> | Metrics | Methods | I2P-Sexual(N=3) | I2P-Sexual(N=10) | I2P-Sexual(N=20) | NSFW-56K(N=3) | NSFW-56K(N=10) | NSFW-56K(N=20) |
> | :--- | :--- | :---: | :---: | :---: | :---: | :---: | :---: |
> | **SSR-N** | NCD | **1.61** | 6.23 | 9.45 | 5.43 | 14.79 | 20.22 |
> | | **NCD-HA** | 1.72 | **6.12** | **9.28** | **4.83** | **13.88** | **20.05** |
> | **ASR** | NCD | 0.61 | 1.01 | 0.97 | 1.98 | 2.26 | 2.08 |
> | | **NCD-HA** | **0.61** | **0.98** | **0.97** | **1.74** | **1.95** | **2.07** |
>
> ---
>
>   ***Q3: Is there a theoretical upper limit to performance (36.1% failure on MMA-Diffusion), and can increasing training seeds or adversarial seed sampling help?***
>
> We fully understand your question. The remaining 36.1% failure rate on the MMA-Diffusion benchmark is primarily due to a sampling efficiency gap, as the continuous, adversarial latent space must be approximated using finite samples. For possible solutions, while increasing the number of random seeds ($N$) yields diminishing returns, we agree that adversarial seed sampling may be an effective approach to break this limit, which has been verified in **Q2** to some extent.

---

> ### Author Response · Authors · 2025-12-03
> **Response to Reviewer cxrR (2/2)**
>
> ***To Reviewer cxrR (2/2):***
>
> ***Q4: Why were multi-response DPO extensions like IPO and SimPO excluded from comparisons?***
>
> Thanks for this suggestion! However, though DPO has been successfully adapted to diffusion models (e.g., via the ELBO formulation), extensions like IPO and SimPO are mathematically ill-suited for the diffusion models due to intrinsic formulation conflicts (this is also why we did not include them as comparsions):
>
> - **Objective Mismatch (IPO):** DPO is derived as a robust classification loss (sigmoid) on the difference in denoising errors, which suits continuous optimization. IPO, conversely, employs a regression loss (squared error) on the exact magnitude of the log-likelihood ratio. Regressing the exact difference in noise prediction errors is highly prone to optimization instability and collapse in high-dimensional continuous space.
>
> - **Structural Incompatibility (SimPO):** SimPO introduces two critical structural conflicts:
>
>   * **Reference-Free Design:** SimPO removes the reference model ($\pi_{ref}$), eliminating the essential anchor required for stable fine-tuning in Diffusion DPO. This risks severe mode collapse as the model loses its guidance for preserving image fidelity.
>
>   * **Topology Conflict:** SimPO relies on sequence length normalization, which is topologically meaningless for fixed-dimension image tensors, breaking the mathematical intent of the loss function.
>
> ---
>
> ***Q5: Will you commit to publishing a safe subset of NCD-10K to facilitate community research?***
>
>  Yes, we are fully committed to releasing a safe, carefully curated subset of NCD-10K (with potentially sensitive or copyright-concerning entries removed) to facilitate reproducible community research, and we will make it publicly available as soon as the reviewing and cleaning process is finalized.

---

### Note · Authors · 2026-01-30

I have read and agree with the venue's withdrawal policy on behalf of myself and my co-authors.

---

### Meta-Review · Area_Chair_7VML · 2026-01-05

**Summary:**

This paper tackles "cross-seed instability" in T2I diffusion models, proposing Noise Contrastive Diffusion (NCD) to prevent harmful outputs across different random seeds. While the problem is practically relevant, the consensus among reviewers is that the contribution is incremental. NCD is viewed primarily as a variation of existing alignment techniques (like DPO/NCA) rather than a distinct algorithmic breakthrough. Even with the additional experiments provided in the rebuttal, the method appears to be a "brute-force" strategy, rather than a theoretical solution that fundamentally resolves the probabilistic failure modes of diffusion models.

**Reviewer Concerns:**

**Addressed**

- Missing Baselines: The authors successfully added comparisons against SOTA methods like AdvUnlearn and DUO.

- Dataset Bias: Evaluations were expanded to include long-tail harmful categories (e.g., violence, self-harm), addressing concerns about the initial over-reliance on sexual content.

**Outstanding**

- Limited Novelty: The core mechanism is strikingly similar to DPO. The primary differentiator—introducing multi-candidate samples—is seen as an engineering tweak rather than a novel algorithmic contribution.

- Theoretical Limitations (Reviewer DEWM): Given the probabilistic nature of diffusion, NCD reduces attack success rates but fails to offer a theoretical guarantee for true "seed invariance." The fact that Seed Success Rate (SSR) still rises with N suggests the instability is suppressed but not rooted out.

- Methodological Overhead: Re-introducing pairwise data processing adds complexity and heavy reliance on offline data quality, diverting from the goal of an elegant algorithmic solution.

**Reviewer Scores:**

Reviewer dHZV: 2 -> 4

Reviewer AdHF: 4

Reviewer DEWM: 4

Reviewer cxrR: 6

---

### Decision · Program_Chairs · 2026-01-26

Reject